# SERQ: Saliency-Aware Low-Rank Error Reconstruction for LLM Quantization

**Yeonsik Park**
Kyung Hee University
plsik125@khu.ac.kr

**Hyeonseong Kim**
Kyung Hee University
hsyoun2222@khu.ac.kr

**Seungkyu Choi**
Yonsei University
seungkc@yonsei.ac.kr

## Abstract

Post-training quantization (PTQ) has emerged as a prevailing technique for deploying large language models (LLMs) efficiently in terms of both memory and computation, across edge devices and server platforms. Existing PTQ methods primarily aim to reduce precision in weights and activations by mitigating quantization errors caused by channel-wise outlier activations (e.g., pre-quantization scaling, online transformations, or low-rank error reconstruction). Among these approaches, error reconstruction with low-rank adaptation (LoRA) has proven particularly effective, as it introduces a lightweight auxiliary computation path without requiring heavy optimization or additional online layers. However, prior studies reveal severe accuracy degradation under W4A4 settings, and conventional low-rank adaptations rely on two sequential factors, necessitating intermediate quantization during inference and thereby limiting low-precision efficiency. In this work, we propose SERQ, a saliency-aware error reconstruction method for low-bit LLM inference that employs a single low-rank compensation matrix. SERQ preserves efficient 4-bit matrix multiplication in linear layers by jointly mitigating quantization errors arising from both activation and weight saliency through three stages: (1) static activation flattening, (2) saliency-aware error reconstruction, and (3) offline weight permutation. The method incurs additional computation only for low-rank error reconstruction via a single decomposition, while all other operations are performed offline, thereby keeping latency overhead minimal. Empirically, SERQ outperforms prior error reconstruction methods under both W4A8 and W4A4 settings, and achieves higher accuracy than state-of-the-art rotation-based W4A4 approaches, while substantially reducing calibration complexity. Code is available at https://github.com/acalabys/SERQ

## 1 Introduction

The demand for efficient deployment of large language models (LLMs) has been rapidly increasing across both server and edge platforms. Quantization has emerged as one of the most effective approaches to reduce the substantial memory and computational costs associated with LLM inference. In particular, post-training quantization (PTQ) techniques (Nagel et al. (2021)) enable low-precision representations of weights and activations involved in large-scale computations, thereby avoiding expensive fine-tuning while maintaining competitive performance.

A central challenge in minimizing quantization errors for LLMs lies in addressing outlier activations across channels. To alleviate this issue, several distribution-flattening approaches have been proposed, including pre-quantization scaling methods (Xiao et al. (2024); Shao et al. (2024)) and online transformation-based techniques that leverage random Hadamard or learned transformations (Ashkboos et al. (2024); Liu et al. (2025)). While recent rotation transformation methods have demonstrated effectiveness in enabling 4-bit integer (INT4) quantization, they typically rely on computationally expensive calibration procedures or suffer from performance variability induced by random matrices, thereby limiting their practicality in general deployment.

An alternative strategy for mitigating activation outliers is matrix decomposition. Recent advances have introduced low-rank error reconstruction methods that integrate quantization with low-rank adaptation (LoRA) (Dettmers et al. (2023); Saha et al. (2024); Zhang et al. (2024); Zhao et al.

(2024)). These approaches leverage low-rank decompositions in matrix multiplication to reduce quantization error by compensating for it through separate low-rank factors. For example, $L^2QER$ (Zhang et al. (2024)) introduces a fully quantized path for low-rank error reconstruction, yielding near-zero accuracy loss under the 4-bit weight, 8-bit activation (W4A8) configuration. Despite their superior adaptability, these methods have not yet achieved W4A4 quantization without noticeable performance degradation. Furthermore, they remain unsuitable for fully low-precision execution, as decomposed matrices are multiplied sequentially, producing intermediate values, requiring an additional on-the-fly quantization process.

In this work, we propose SERQ, a saliency-aware error reconstruction method that enables low-precision LLM inference (e.g., W4A4, W4A8) using a single low-rank decomposition. Unlike standard low-rank approximations that rely on two low-rank factors, our method unifies error correction into a single matrix by jointly addressing activation and weight saliency. This design avoids the overhead of an additional sequential low-rank branch during inference, while effectively mitigating quantization errors arising from both activation outliers and salient weights. As a result, SERQ achieves more efficient low-precision inference while maintaining high accuracy, outperforming prior quantization approaches in the challenging W4A4 precision setting.

To the best of our knowledge, this is the first work to realize 4-bit matrix multiplication in linear layers by employing low-rank error reconstruction, a method recognized for its adaptability and minimal calibration overhead. Following this principle, SERQ introduces no additional layers for online processing and avoids costly calibration procedures such as hyperparameter search, or other compute-intensive training operations. Our contributions are summarized as follows.

- We propose SERQ, a novel W4A4 quantization scheme for LLMs that employs a single saliency-guided low-rank matrix for accurate error reconstruction. Our method operates in three steps: static activation flattening, saliency-aware error reconstruction, and offline weight permutation.
- SERQ enables 4-bit matrix multiplication (e.g. INT4, MXFP4) in linear layers, thereby minimizing inference overhead in low-rank computation. Moreover, the proposed flattening and permutation schemes are merged into weight parameters and preprocessed in adjacent layers, allowing them to be fully managed offline with no additional latency.
- We validate our scheme across various LLMs with comprehensive evaluations. Compared to prior LoRA-based methods, our approach achieves superior performance in both W4A8 and W4A4 configurations while using only a single low-rank matrix. Furthermore, we compare against state-of-the-art rotation-based W4A4 quantization approaches, demonstrating superior accuracy while significantly reducing calibration complexity.

## 2 BACKGROUND AND MOTIVATION

### 2.1 LLM QUANTIZATION

Quantization maps high-precision values to low-precision representations, improving both memory and compute efficiency. The basic integer quantization with max-scaling can be expressed as:

$$\boldsymbol{X}_q = \text{clip}(\lceil \boldsymbol{X}/\boldsymbol{s} \rfloor), \quad \boldsymbol{s} = \max(|\boldsymbol{X}|)/(2^{n-1} - 1), \quad \hat{\boldsymbol{X}} = \boldsymbol{s} \cdot \boldsymbol{X}_q \tag{1}$$

where $s$ is the scale factor, $n$ is the bit-width, $\lceil \cdot \rfloor$ denotes round-to-nearest, and $\text{clip}()$ clamps values to $[-2^{n-1} - 1, \ 2^{n-1} - 1]$. Reducing bit-width provides near-linear storage compression, alleviating the growing memory demands of model parameters and KV-cache. At runtime, quantization also reduces memory traffic, thereby improving bandwidth efficiency. When activations are quantized in addition to weights, the core linear operation can be executed as an integer GEMM, further accelerating inference latency and throughput. For a transformer linear projection $\boldsymbol{y} = \boldsymbol{W}\boldsymbol{x}$,

$$\boldsymbol{y} \approx \boldsymbol{s}_W(\boldsymbol{W}_q\boldsymbol{X}_q)\boldsymbol{s}_X \tag{2}$$

As widely recognized, the accuracy of LLM quantization for both weights and activations critically depends on how activation outliers are handled. Numerous approaches have been proposed to mitigate the quantization error introduced by such outliers. Early methods, such as SmoothQuant (Xiao

et al. (2024)) and OmniQuant (Shao et al. (2024)), employ distribution-flattening techniques that balance activations and weights through pre-quantization scaling. Another line of work decomposes matrix multiplication by assigning a separate high-precision path for outliers, as in `LLM.int8()` (Dettmers et al. (2022)) and QUIK (Ashkboos et al. (2023)). However, these primitive techniques exhibit significant accuracy degradation when applied to sub-8-bit LLMs.

More recently, online transformation techniques that rotate tensors to flatten the distribution have demonstrated effectiveness for 4-bit quantization with minimal latency overhead. Quarot (Ashkboos et al. (2024)) applies random Hadamard transformations, while SpinQuant (Liu et al. (2025)) learns rotation matrices to suppress outliers and reduce the performance variance. While effective in the W4A4 setting, existing methods still incur notable accuracy loss and either suffer from high variance due to random Hadamard matrices or require costly training to optimize transformation matrices, limiting deployment practicality. Meanwhile, LoRA-based LLM quantization has emerged as a powerful approach for mitigating quantization errors. In a similar vein, error reconstruction with low-rank factors via matrix decomposition has proven particularly effective, while avoiding heavy optimization or additional online layers. Further details are discussed in section 2.2.

## 2.2 Low-Rank Error Reconstruction

Recent advances in LoRA for parameter-efficient fine-tuning of foundation LLMs suggest its potential for applicability to LLM quantization as well. LoRA has been adapted to compensate for quantization errors by leveraging the auxiliary path of low-rank matrices, a strategy we refer to as low-rank error reconstruction. This approach introduces a LoRA-style compensator that restores the principal quantization error without requiring heavy retraining or complex deployment modifications. Concretely, it corrects the quantization error of a full-precision weight matrix $W^{d \times d}$, where d denotes the hidden dimension of a layer, by augmenting its low-bit proxy with low-rank factors.

$$\hat{W} = Q(W) + L_1 L_2, \quad L_1 L_2 \approx W - Q(W) \tag{3}$$

Where $Q(W)$ is the quantized weight, $L_1 \in \mathbb{R}^{d \times r}$, $L_2 \in \mathbb{R}^{r \times d}$, and $r \ll d$ is the rank. The parameter and compute overhead of the low-rank factors is only $2rd$; with commonly used ranks $r \in \{32, 64\}$ (e.g., $d = 4096$), this amounts to about 1.6-3.1%. Moreover, the LoRA-path is simple and lightweight, which makes it highly applicable across diverse model architectures and hardware platforms. Consequently, it has emerged as a prominent and widely adopted quantization method.

**Gradient-based Methods.** Gradient-based methods compensate the quantization errors by leveraging loss gradients on a small calibration set to learn low-rank factors, typically instantiating a compensator $L_1 L_2$ that reduces mismatch between the full-precision model and its quantized counterpart. QLoRA (Dettmers et al. (2023)) advanced this line of work by fine-tuning LoRA adapters on top of 4-bit weights, recovering task performance with modest resource overhead. At sub-4-bit precision, methods such as LQ-LoRA (Guo et al. (2024)), LoftQ (Li et al. (2023)) and QA-LoRA (Xu et al. (2023)) consistently mitigate degradation and sustain competitive accuracy, highlighting the viability of ultra-low-bit quantization.

**SVD-based Methods.** Another approach is to reconstruct quantization errors by low-rank matrices obtained via singular-value decomposition (SVD). This approach forms the compensator $L_1 L_2$ from the quantization error $E$ by exploiting its rapidly decaying singular spectrum, allowing the dominant error to be captured with compact low-rank factors. Recent methods such as ZeroQuant-v2 (Yao et al. (2023)), CALDERA (Saha et al. (2024)), and $L^2$QER (Zhang et al. (2024)) adopt this SVD-based optimization to extract low-rank factors in a training-free and lightweight calibration process. In particular, $L^2$QER formalizes weight–activation quantization using SVD-based error reconstruction, preserving integer matrix multiplications across both the main and low-rank branches.

## 2.3 Deployment of Low-Rank Error Reconstruction

In low-rank error reconstruction, a linear layer introduces an additional computation path for the low-rank branch, which cannot be parallelized cleanly with the main branch. Consider the case where both weights and activations are quantized. To enable the main branch to leverage low-precision matrix multiplication (e.g., INT4 or MXFP4 GEMM kernels), the LoRA-augmented linear layer must be evaluated in its decomposed form, such as $X_q(W_q + L_1 L_2) = X_q W_q + X_q L_1 L_2$.

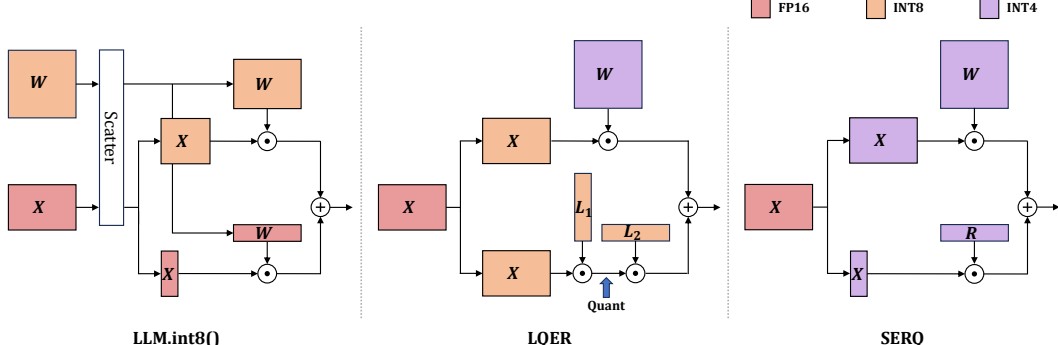

Figure 1: Computation flow of a linear layer under different matrix decomposition methods. `LLM.int8()` employs a mixed-precision scheme of INT8 and FP16 by assigning separate computation paths for outliers and non-outliers. $L^2$QER applies SVD-based error reconstruction with a mixed-precision path of INT4 and INT8. In contrast, the proposed SERQ leverages a saliency-guided low-rank matrix and provides a unified computation path with INT4 or MXFP4 precision.

This requires two sequential matrix multiplications with low-rank matrices. Although each operation is relatively inexpensive, the overhead becomes non-negligible when the main branch benefits from highly optimized low-precision GEMM kernels. To address this, $L^2$QER (Zhang et al. (2024)) quantizes the low-rank matrices as well, allowing the entire computation to proceed in a fully quantized path, as expressed in $\boldsymbol{X}_q\boldsymbol{W}_q + \mathrm{Q}(\boldsymbol{X}_q\boldsymbol{L}_{1,q})\boldsymbol{L}_{2,q}$. However, this formulation introduces an additional on-the-fly quantization step for intermediate results between the sequential multiplications, resulting in inefficiency for low-precision deployment.

Figure 1 shows the overall datapath of prior approaches compared to ours, all based on decomposed matrix multiplication. Prior to error reconstruction methods, `LLM.int8()` (Dettmers et al. (2022)) handled outliers via a separate high-precision path, but this required on-the-fly scattering and FP16 computation, introducing substantial latency. An error reconstruction method $L^2$QER (Zhang et al. (2024)) later achieved a fully quantized datapath with mixed precision, yet still relied on on-the-fly quantization and two additional narrow matrix multiplications, resulting in latency and degraded accuracy in the W4A4 setting. In contrast, we propose SERQ, a novel error-reconstruction method that employs a single low-rank matrix, eliminating on-the-fly quantization and enabling a fully 4-bit end-to-end computation path. Further details are provided in section 3.

## 3 METHOD

We present SERQ, a saliency-aware error reconstruction method that jointly accounts for weight and activation saliency within a single low-rank matrix. We first show that saliency plays a central role in compensating weight-side quantization error, motivating a saliency-guided low-rank design. We then detail how SERQ incorporates activation saliency into the weight matrix, enabling error reconstruction based on integrated weight saliency while minimally impacting inference latency.

### 3.1 SALIENCY-AWARE LOW-RANK ADAPTATION

Prior error-reconstruction methods typically approximate the full-weight quantization error $\boldsymbol{E} = \boldsymbol{W} - \mathrm{Q}(\boldsymbol{W})$ using a truncated SVD of the weight matrix, $\boldsymbol{E} \approx \boldsymbol{U}_r\Sigma_r\boldsymbol{V}_r^{\mathrm{T}}$, where $\boldsymbol{U}_r \in \mathbb{R}^{d\times r}$ and $\boldsymbol{V}_r^{\mathrm{T}} \in \mathbb{R}^{r\times d}$ contain the top-$r$ singular vectors and $\Sigma_r \in \mathbb{R}^{r\times r}$ is the diagonal matrix of the corresponding singular values. While effective, this global decomposition overlooks where the error actually concentrates: the fixed rank budget is distributed across all rows and columns, diluting capacity on the most problematic regions (e.g., salient weights). In addition, quantizing the low-rank factors themselves introduces further loss, undermining the efficiency of error reconstruction.

Building on the theoretical insight that quantization errors vary across weight rows in linear operations, AWQ (Lin et al. (2024b)) demonstrates that protecting only a small fraction ($\sim$1%) of salient channels (i.e., their corresponding weight rows) using the activation distribution can substantially reduce the error. Motivated by this, we evaluate error reconstruction using SVD with a fixed rank

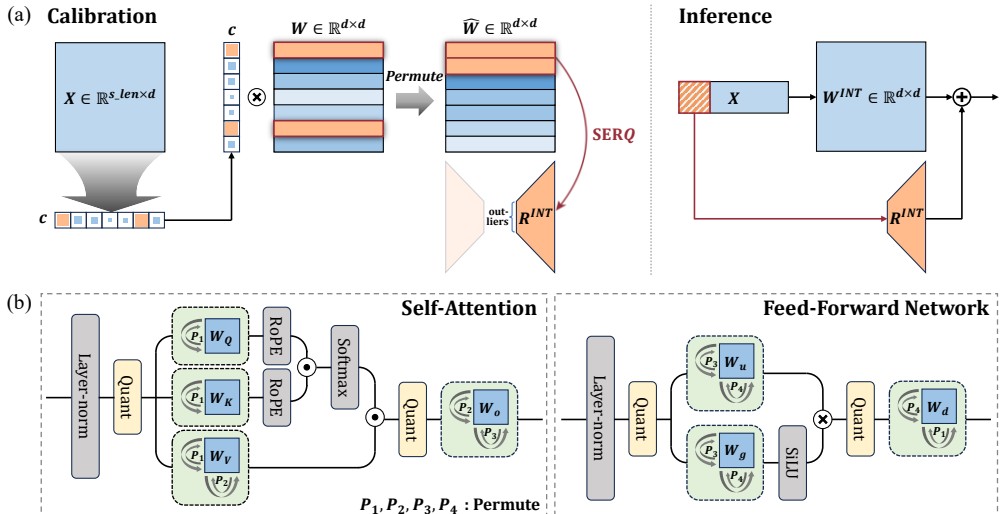

Figure 2: (a) Overall SERQ implementation. During calibration, saliency rows are determined via activation scaling, followed by weight row permutation. During inference, error reconstruction is performed through a residual path computed only on the salient components, alongside the main path. (b) Computation flow of a decoder layer. The merged row- and column-wise weight permutation enables offline preprocessing of both current weight rows and subsequent activation channels.

size but restricted to salient rows selected by activation scales (see Appendix A.1). We find that selecting only a small number of salient rows for SVD improves perplexity compared to using the entire matrix. In other words, rather than extracting ranks from the full weight matrix via SVD, we directly identify rank candidates based on row-wise saliency, allowing to reconstruct quantization error only on the most influential data. This method requires only a narrowed matrix from these salient rows, equal to the rank size, ensuring high-fidelity approximation while enabling a single matrix decomposition for the low-rank branch. In section 3.2, we describe how to integrate activation statistics into the weight matrix to determine salient rows that disproportionately affect linear operations, and how to jointly reconstruct the error to improve accuracy using a full low-precision computing path.

## 3.2 IMPLEMENTATION

**Static Activation Flattening.** Activation quantization is notoriously fragile due to channel-wise outliers. Existing methods often rely on online outlier-handling techniques (e.g., rotation transforms or auxiliary layers), which are effective but introduce additional latency. Instead, since our method provides a residual path for error reconstruction at the linear layer, we avoid the online flattening process and revisit SmoothQuant (Xiao et al. (2024)), which flattens activation distributions using static per-channel scaling. Specifically, activations are scaled by a factor $s$, and the corresponding scale is folded into the weights. The operation in a linear layer can therefore be expressed as:

$$Y = XW = (X \cdot \mathrm{diag}(s^{-1}))(\mathrm{diag}(s) \cdot W) = \widetilde{X}\widetilde{W} \tag{4}$$

The scaling factors are obtained during calibration and merged into adjacent layers offline, incurring no runtime overhead. While this strategy alleviates outliers, it shifts the quantization burden onto the weights, increasing the difficulty of weight quantization. However, unlike standalone prior approaches, the combined use of low-rank reconstruction enables effective compensation for the induced weight errors. Concretely, SERQ identifies salient weights after the flattening step and restores the residuals using a single low-rank matrix.

**Saliency-Aware Error Reconstruction.** The per-channel static flattening process pushes the scale of activation outliers into the corresponding weight rows. Assuming that the original weights follow a normal distribution, the salient rows in the folded weights can be identified directly by their scales. These rows subsequently accumulate significant errors when repeatedly multiplied with the activation matrix. To mitigate this, we introduce a low-rank compensator matrix $R \in \mathbb{R}^{r \times d}$ that

corrects the quantization errors in the $r$ salient weight rows, denoted $\widetilde{\boldsymbol{W}}_s$. Considering the weight rows permuted in descending order of saliency ($P$), the folded matrix $\widehat{\boldsymbol{W}}$ and the saliency-aware low-rank matrix $\boldsymbol{R}$ can be defined as:

$$\widehat{\boldsymbol{W}} = P \cdot \mathrm{diag}(\boldsymbol{s}) \cdot \boldsymbol{W} = P \cdot \widetilde{\boldsymbol{W}} = [\widetilde{\boldsymbol{W}}_s; \widetilde{\boldsymbol{W}}_r], \quad \boldsymbol{R} = \widetilde{\boldsymbol{W}}_s - \mathrm{Q}(\widetilde{\boldsymbol{W}}_s) \tag{5}$$

Here, $\boldsymbol{W}_r$ denotes the remaining weight rows. As shown, the residual errors of the salient rows are captured by the low-rank matrix to be used for reconstruction during matrix multiplication via an additional path. Then the overall linear operation can be described as:

$$\boldsymbol{Y} = (\boldsymbol{X} \cdot \mathrm{diag}(\boldsymbol{s}^{-1}) \cdot \boldsymbol{P}^{-1})(\boldsymbol{P} \cdot \mathrm{diag}(\boldsymbol{s}) \cdot \boldsymbol{W}) = \widehat{\boldsymbol{X}}\widehat{\boldsymbol{W}} \tag{6}$$

$$\mathrm{Q}(\widehat{\boldsymbol{X}}) \cdot \mathrm{Q}(\widehat{\boldsymbol{W}}) = \mathrm{Q}([\widetilde{\boldsymbol{X}}_s \ \widetilde{\boldsymbol{X}}_r]) \cdot \mathrm{Q}([\widetilde{\boldsymbol{W}}_s; \widetilde{\boldsymbol{W}}_r]) + \mathrm{Q}(\widetilde{\boldsymbol{X}}_s) \cdot \boldsymbol{R}$$
$$\approx \widehat{\boldsymbol{X}_q} \cdot \widehat{\boldsymbol{W}_q} + \widehat{\boldsymbol{X}_{s,q}} \cdot \mathrm{Q}(\boldsymbol{R}) \tag{7}$$

Importantly, we quantize the low-rank matrix as well, enabling pure low-precision computation across the entire path. Unlike SVD, directly extracting salient rows requires only a single matrix multiplication in the residual path, thereby eliminating intermediate quantization. Furthermore, only the activation channels corresponding to the salient rows ($\widetilde{\boldsymbol{X}}_s \in \mathbb{R}^{s\_len \times r}$) participate in the residual operation, enabling a computation-efficient low-rank multiplication of $\mathbb{R}^{s \times r} \times \mathbb{R}^{r \times d}$. The overall SERQ process, including both calibration and inference, is illustrated in Figure 1(a).

**Offline Weight Permutation.** We have aforementioned that the weights and activations must be properly reordered based on their saliency (e.g. $\widehat{\boldsymbol{X}} = [\widetilde{\boldsymbol{X}}_s \ \widetilde{\boldsymbol{X}}_r], \widehat{\boldsymbol{W}} = [\widetilde{\boldsymbol{W}}_s; \widetilde{\boldsymbol{W}}_r]$). To address this, we propose a mergeable weight permutation scheme that eliminates latency overhead during inference. Both the rows and columns of the weight matrix are permuted offline according to the saliency order, enabling matrix multiplication to be executed directly on the appropriately reordered weights and activations. Figure 2(b) illustrates the computation flow of a single decoder layer, highlighting the offline permutation step. Based on saliency levels determined during calibration, row-wise permutations are prearranged for all weight parameters. The corresponding activations must follow the same channel order, which can be achieved by applying column-wise permutations to the preceding layer's weight matrix. For example, the permutation order $P_4$ of the down-projection layer can be propagated to the weight columns of the preceding up- and gate-projection layers, ensuring that the activation outputs are produced in the desired order. Consequently, the resulting activations naturally align with $P_4$, allowing the down-projection to operate without additional reordering. In this way, all linear layers avoid on-the-fly reordering, and inference proceeds without latency overhead.

## 4 EXPERIMENTS

### 4.1 SETTINGS

**Models and Tasks.** We conduct experiments on LLaMA-2 (7B, 13B and 70B; Touvron et al. (2023)), LLaMA-3 (3.1 8B and 3.2 1B/3B; Grattafiori et al. (2024)), and Qwen-2.5 3B (Qwen et al. (2025)) models. Our evaluation covers eight zero-shot commonsense reasoning tasks—PIQA (Bisk et al. (2019)), SIQA (Sap et al. (2019)), ARC-Easy/Challenge (Clark et al. (2018)), HellaSwag (Zellers et al. (2019)), Winogrande (Sakaguchi et al. (2021)), BoolQ (Clark et al. (2019)), and OpenBookQA (Mihaylov et al. (2018))—and also provides perplexity scores on the WikiText2 test set (Merity et al. (2016)) as well as the MMLU benchmark (Hendrycks et al. (2021)). We further report results on generation tasks using the GSM8K (Cobbe et al. (2021)) and LongBench (Bai et al. (2024)) datasets.

**Implementation Details.** The calibration set is constructed from 128 random samples of the WikiText-2 dataset (Merity et al. (2016)) to identify salient rows. SERQ uses a rank size of 128, which is equivalent in effective bit width to using two low-rank matrices of rank 64. For quantization, the group size is set to 128, and weights are quantized using either GPTQ (Frantar et al. (2023)) or round-to-nearest (RTN) symmetric integer quantization, while activations are quantized with RTN asymmetric integer quantization. For the W4A4 configuration, we also provide the standard 4-bit Microscaling (MX) format, MXFP4, alongside the integer format, to demonstrate its implementation

Table 1: Comparison with matrix decomposition methods. We compare perplexity scores, average zero-shot common sense reasoning accuracy, and average MMLU accuracy. Results under different precision settings are obtained by modifying their publicly released codebase (See Appendix A.7).

| #Bits | Method | #Eff. ($w$) | LLaMA-2 7B | | | LLaMA-2 70B | | | LLaMA-3 8B | | |
|---|---|---|---|---|---|---|---|---|---|---|---|
| | | | PPL(↓) | 0-shot(↑) | MMLU(↑) | PPL(↓) | 0-shot(↑) | MMLU(↑) | PPL(↓) | 0-shot(↑) | MMLU(↑) |
| FP16 | baseline | 16 | 5.47 | 64.09 | 41.83 | 3.53 | 70.54 | 65.44 | 6.13 | 67.16 | 62.13 |
| W4A8 | LLM.int4() | -† | 7.28 | 59.64 | 29.01 | 7.41e+3 | 49.71 | 43.56 | 10.82 | 60.86 | 46.72 |
| | L²QER | 4.35 | 5.83 | 63.35 | 39.4 | 3.55 | 69.78 | 64.45 | 7.16 | 65.68 | 57.81 |
| | SERQ (RTN) | 4.24 | 5.64 | **63.41** | 40.21 | 3.44 | **70.35** | 64.45 | 6.71 | **66.4** | 58.49 |
| | SERQ (GPTQ) | 4.24 | **5.59** | 63.04 | **40.29** | **3.43** | 70.31 | **64.62** | **6.52** | 66.23 | **60.25** |
| W4A4 | LLM.int4() | -† | 6.32e+2 | 34.85 | 24.41 | 2.76e+4 | 39.77 | 23.67 | 4.87e+2 | 36.12 | 23.61 |
| | L²QER | 4.24 | 7.37 | 57.67 | 29.63 | 4.55 | 66.92 | 56.44 | 11.44 | 55.44 | 38.33 |
| | L²QER-MXFP4 | 4.37 | 6.3 | 60.95 | 35.22 | 3.81 | 69.03 | 61.48 | 7.83 | **63.33** | **53.82** |
| | SERQ (RTN) | 4.24 | 6.03 | 61.77 | **38.03** | 3.65 | **69.68** | **63.15** | 8.07 | 62.49 | 51.84 |
| | SERQ (GPTQ) | 4.24 | **5.97** | **61.87** | 37.03 | **3.64** | 69.38 | 63.09 | 7.75 | 62.41 | 53.8 |
| | SERQ-MXFP4 | 4.37 | 6.22 | 61.26 | 35.25 | 3.79 | 69.58 | 61.64 | **7.63** | 62.71 | 53.48 |

| #Bits | Method | #Eff. ($w$) | LLaMA-3.2 1B | | | LLaMA-3.2 3B | | | Qwen-2.5-3B | | |
|---|---|---|---|---|---|---|---|---|---|---|---|
| | | | PPL(↓) | 0-shot(↑) | MMLU(↑) | PPL(↓) | 0-shot(↑) | MMLU(↑) | PPL(↓) | 0-shot(↑) | MMLU(↑) |
| FP16 | baseline | 16 | 9.75 | 54.82 | 36.76 | 7.81 | 62.66 | 54.06 | 8.03 | 63.82 | 65.12 |
| W4A8 | LLM.int4() | -† | 32.39 | 44.56 | 24.01 | 12.47 | 56.06 | 41.48 | 9.98e+5 | 35.59 | 25.52 |
| | L²QER | 4.35 | 12.06 | 50.97 | 31.23 | 8.78 | 60.36 | 51.24 | 12.25 | 54.71 | 26 |
| | SERQ (RTN) | 4.24 | 11.14 | 52.76 | **32.77** | 8.38 | 61.27 | **51.45** | 8.67 | **63.78** | 62.79 |
| | SERQ (GPTQ) | 4.24 | **10.45** | **53.93** | 27.1 | **8.18** | **61.54** | 50.24 | **8.36** | 63.7 | **63.25** |
| W4A4 | LLM.int4() | -† | 1.7e+3 | 35.19 | 24.11 | 4.37e+2 | 35.77 | 23.5 | 8.51e+5 | 36.67 | 22.97 |
| | L²QER | 4.24 | 30.83 | 42.17 | 25.67 | 14.11 | 50.92 | 32.25 | 221.5 | 38.37 | 23.81 |
| | L²QER-MXFP4 | 4.37 | 13.78 | 50.22 | 27.1 | 9.59 | 58.41 | **46.85** | 10.46 | 60.56 | 55.87 |
| | SERQ (RTN) | 4.24 | 13.57 | 50.15 | **29.11** | 9.43 | 58.4 | 46.36 | 9.66 | **61.24** | 59.52 |
| | SERQ (GPTQ) | 4.24 | **12.52** | **50.44** | 26.34 | 9.15 | **58.5** | 45.7 | **9.35** | 60.78 | **59.55** |
| | SERQ-MXFP4 | 4.37 | 13.71 | 49.94 | 27.23 | **8.74** | 57.5 | 41.33 | 9.79 | 60.43 | 56.13 |

† In the case of LLM.int4(), the effective bit-width varies dynamically.

and measured speed on an NVIDIA RTX PRO 6000 GPU with Blackwell architecture support for MX format kernels (Rouhani et al. (2023)) (See Appendix A.2.2).

**Compared Methods.** We primarily compare our method against the state-of-the-art error reconstruction approach L²QER under both W4A4 and W4A8 configurations. As a baseline for matrix decomposition with low-rank error reconstruction, we include LLM.int4(), the 4-bit variant of LLM.int8() (Dettmers et al. (2022)). We further compare against state-of-the-art W4A4 quantization methods that employ distribution-flattening techniques, including the rotation-based approaches Quarot Ashkboos et al. (2024) and SpinQuant Liu et al. (2025).

## 4.2 EVALUATION RESULTS

**Comparison with Low-Rank Matrix Decomposition Methods.** We evaluate the W4A8 and W4A4 precision configurations to demonstrate the effectiveness of SERQ in low-rank error reconstruction. Both L²QER and SERQ are compared under the same group size, with low-rank matrix dimensions matched to ensure the same parameter counts. The difference in effective weight bit-width under W4A8 arises because L²QER employs 8-bit precision for its low-rank matrices, whereas SERQ preserves 4-bit precision. As shown in Table 1, SERQ consistently outperforms LLM.int4() and L²QER across most tasks, while achieving the lowest effective bit-width. Notably, SERQ is compatible with both RTN and GPTQ for weight quantization. Details of applying GPTQ robustly to SERQ are provided in Appendix A.3.

The accuracy gap is more pronounced under the W4A4 setting. While prior INT4 quantization methods suffer from severe degradation in most cases, SERQ maintains high accuracy across all tasks. L²QER performs especially poorly on LLaMA-3 models, failing to preserve accuracy when both activations and low-rank matrices are quantized to 4-bit. We further provide MXFP4 implementations with the default group size of 32. Although MXFP4 proves adequate for both methods, the sequential reconstruction path in L²QER, which requires two low-rank matrices, introduces significant latency overhead, which is examined in detail in the GPU performance analysis.

**Comparison with W4A4 Distribution Flattening Methods.** Rotation-based methods are recognized as state-of-the-art W4A4 quantization approaches, achieving the lowest effective bit-width

Table 2: Comparison with W4A4 distribution flattening methods. Latency overhead is measured as the additional computation time per linear layer relative to 4-bit GEMM (See Appendix A.7).

| #Bits | Method | #Eff. ($w$-bits) | Training-free | Latency overhead | LLaMA-2 7B | | | LLaMA-2 13B | | |
|---|---|---|---|---|---|---|---|---|---|---|
| | | | | | PPL($\downarrow$) | 0-shot($\uparrow$) | MMLU($\uparrow$) | PPL($\downarrow$) | 0-shot($\uparrow$) | MMLU($\uparrow$) |
| FP16 | baseline | 16 | ✓ | 81.4% | 5.47 | 64.09 | 41.83 | 4.88 | 66.53 | 52.04 |
| W4A4 | SmoothQ | ~4 | ✓ | ✗ | 1.51e+4 | 35.44 | 26.04 | 1.32e+4 | 34.53 | 23.85 |
| | SmoothQ(g128) | 4.13 | ✓ | ✗ | 7.49 | 57.15 | 30.4 | 6.31 | 61.28 | 39.83 |
| | QuaRot | ~4 | ✓ | 19.8% | 6.15 | 59.53 | 33.58 | 5.41 | 62.55 | 47.25 |
| | SpinQuant | ~4 | ✗ | 19.8% | 6.0 | 61 | 34.8 | **5.2** | 64.8 | **47.8** |
| | SERQ | 4.24 | ✓ | **18.7%** | 5.97 | **61.87** | **37.03** | 5.2 | **64.82** | 47.17 |

| #Bits | Method | LLaMA-3 8B | | | LLaMA-3.2 1B | | | LLaMA-3.2 3B | | |
|---|---|---|---|---|---|---|---|---|---|---|
| | | PPL($\downarrow$) | 0-shot($\uparrow$) | MMLU($\uparrow$) | PPL($\downarrow$) | 0-shot($\uparrow$) | MMLU($\uparrow$) | PPL($\downarrow$) | 0-shot($\uparrow$) | MMLU($\uparrow$) |
| FP16 | baseline | 6.13 | 67.16 | 62.13 | 9.75 | 54.82 | 36.76 | 7.81 | 62.66 | 54.06 |
| W4A4 | SmoothQ | 4.71e+5 | 36.34 | 25.17 | 1.53e+5 | 35.59 | 24.37 | 3.73e+4 | 35.72 | 23.41 |
| | SmoothQ(g128) | 17.26 | 48.97 | 29.3 | 69.22 | 40.04 | 24.43 | 53.33 | 44.17 | 27.31 |
| | QuaRot | 8.41 | 59.12 | 47.29 | 13.17 | 50.03 | **26.64** | 9.73 | 55.76 | 44.75 |
| | SpinQuant | 8.26 | 61.75 | 49.93 | 13.47 | 48.95 | 26.38 | 10.15 | 56.88 | 42.42 |
| | SERQ | **7.75** | **62.41** | **53.8** | 12.52 | 50.44 | 26.34 | **9.15** | **58.5** | **45.7** |

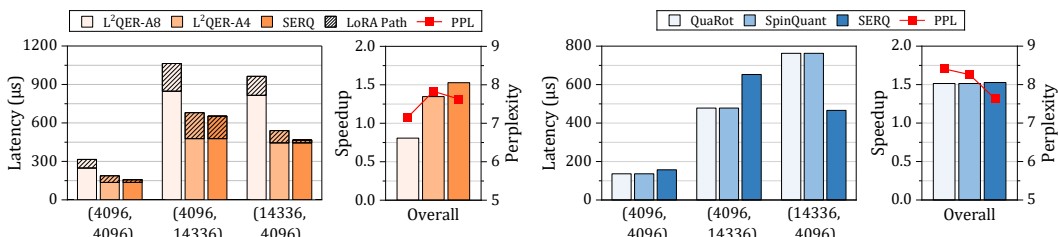

Figure 3: GPU performance comparison. We report latency overhead analysis across various matrix sizes (batch size is 1 and token length is 4k). SERQ is particularly effective for larger row-sized matrices. (See Appendix A.6).

with minimal additional parameters. We therefore compare two representative rotation methods, Quarot and SpinQuant, along with SmoothQuant's distribution-flattening baseline (Xiao et al. (2024)). The reported accuracies for rotation methods are obtained without key-value quantization for fair comparison, using GPTQ without grouping for compatibility with low-precision GPU kernels. As shown in Table 2, SpinQuant generally outperforms Quarot due to its learned rotation matrices, but this advantage diminishes on compact LLaMA-3.2 models, where both methods suffer from significant degradation. In contrast, SERQ achieves consistently higher accuracy, with the improvements most pronounced on LLaMA-3 models. Although SERQ incurs a slightly higher effective bit-width due to the inclusion of low-rank matrices and scaling factors, its per-layer latency overhead is lower than that of rotation-based methods.

**GPU Performance Comparison.** To examine inference performance with the proposed method, we measure execution time using low precision GEMM kernels implemented in the NVIDIA CUT-LASS library (cut (2025)). Since the fifth generation Blackwell Tensor Core architecture does not support INT4 GEMM kernels, while being substantially faster than the previous Ampere generation, we evaluate rotation-based methods in the FP4 format on the same latest core. Mixed precision computation for $L^2$QER with W4A8 is measured with the MXFP mixed-precision kernel, while the Fast Hadamard transform for online rotation is executed using an FP32 kernel (Ashkboos et al. (2024)).

We benchmark the latency of linear operations across different dimension sizes in the LLaMA-3 8B decoder layer, and also report the overall speedup relative to an FP16 GEMM baseline. As shown in the left panel of Figure 3, SERQ outperforms both W4A4 and W4A8 settings of $L^2$QER, while maintaining competitive perplexity. Compared directly with $L^2$QER under W4A4, our single matrix error reconstruction path reduces latency overhead by up to 4.5× compared to the LoRA path, which requires two sequential low rank multiplications, thereby delivering the highest speedup. The right panel of Figure 3 compares performance against rotation-based methods. Since key-value quantization is excluded, the only online rotation occurs between the up or gate projection layer and the down projection layer. However, due to the unbalanced matrix dimensions, rotation introduces a significant overhead, about 1.6× greater than SERQ, which increases the latency of a single decoder

Table 3: End-to-end GPU inference speed and memory measurements for LLaMA-3 8B. We report Time to First Token (TTFT), Time per Output Token (TPOT), and peak memory consumption. The input sequence length is fixed at 2k tokens for all measurements.

| Bsz | Method | TTFT(ms) | Speedup | Peak Mem(GB) | Saving | TPOT(ms) | Speedup |
|---|---|---|---|---|---|---|---|
| 1 | FP16 | 132.38 | ×1 | 17.44 | ×1 | 18.44 | ×1 |
| | MXFP4 | 56.03 | ×2.36 | 6.94 | ×2.51 | 25.48 | ×0.72 |
| | SERQ-MXFP4 | 62.31 | ×2.12 | 7.03 | ×2.48 | 39.28 | ×0.47 |
| 8 | FP16 | 1252.24 | ×1 | 27.9 | ×1 | 61.93 | ×1 |
| | only-MXFP4 | 542.58 | ×2.31 | 17.4 | ×1.6 | 54.49 | ×1.14 |
| | SERQ-MXFP4 | 587.17 | ×2.13 | 17.48 | ×1.6 | 57.08 | ×1.08 |
| 16 | FP16 | 2515.08 | ×1 | 39.85 | ×1 | 114.24 | ×1 |
| | only-MXFP4 | 1113.06 | ×2.26 | 29.35 | ×1.36 | 100.71 | ×1.13 |
| | SERQ-MXFP4 | 1206.69 | ×2.08 | 29.44 | ×1.35 | 103.27 | ×1.11 |
| 32 | FP16 | 4987.05 | ×1 | 63.76 | ×1 | 218.01 | ×1 |
| | only-MXFP4 | 2257.54 | ×2.21 | 53.25 | ×1.2 | 193.7 | ×1.13 |
| | SERQ-MXFP4 | 2453.29 | ×2.03 | 53.34 | ×1.2 | 202.13 | ×1.08 |

Table 4: Effect of rank size on perplexity.

| #Rank | LLaMA-3 8B | LLaMA-3.2 1B | LLaMA-3.2 3B |
|---|---|---|---|
| 0 | 9.8 | 15.09 | 10.21 |
| 16 | 8.28 | 14.32 | 9.71 |
| 32 | 8.24 | 14.09 | 9.59 |
| 64 | 8.18 | 13.97 | 9.57 |
| 128 | 8.07 | 13.57 | 9.43 |
| 256 | 7.98 | 13.25 | 9.32 |

Table 5: Effect of calibration data on perplexity.

| Datasets | #Sample | LLaMA-3 8B | LLaMA-3.2 1B | LLaMA-3.2 3B |
|---|---|---|---|---|
| | 512 | 8.07 | 13.57 | 9.43 |
| Wiki | 128 | 7.98 | 13.6 | 9.43 |
| | 32 | 8.08 | 13.51 | 9.45 |
| | 512 | 7.91 | 13.63 | 9.47 |
| Pile | 128 | 7.96 | 13.54 | 9.44 |
| | 32 | 8.18 | 13.44 | 9.52 |

layer. As a result, SERQ achieves the best perplexity score while delivering comparable speedups to rotation-based methods, with only about one percent additional latency overhead.

**End-to-End GPU Performance.** To further demonstrate the feasibility of deploying SERQ on Blackwell GPUs, we evaluate the end-to-end latency of both the prefill and decoding stages using the LLaMA-3 8B model. We report Time to First Token (TTFT) for the prefill stage and Time per Output Token (TPOT) for decoding, along with peak memory consumption. As shown in Table 3, SERQ consistently achieves more than a 2× speedup across all batch sizes, while incurring less than a 10% latency overhead relative to vanilla MXF4 acceleration, yet delivering substantially improved accuracy. Although single batch decoding exhibits increased latency under quantization, a trend also observed with MXFP4, we observe moderate speedups at larger batch sizes with only marginal latency overhead compared to MXFP4. Furthermore, consistent with the primary objective of quantization, peak memory usage during the prefill stage is reduced by up to 2.48× relative to the FP16 baseline.

### 4.3 ABLATION STUDIES

**Sensitivity on the Rank Size.** SERQ constructs a low rank matrix of size $r$. While we fix the rank size to align with prior methods, we also analyze its impact separately. We evaluate three models, LLaMA-3 8B and LLaMA-3.2 1B and 3B, by varying the rank size, which corresponds to the number of salient rows. As shown in Table 4, perplexity decreases monotonically with larger ranks, indicating improved accuracy. However, the improvement quickly saturates, and even the smallest setting of $r = 16$ (equivalent to rank 8 in LoRA) yields competitive results.

**Sensitivity on Calibration Samples.** Table 5 reports perplexity scores obtained with varying calibration dataset sizes, which are used to determine saliency. We further evaluate our scheme on the Pile dataset Gao et al. (2020) to assess sensitivity to both sample size and dataset choice. The results indicate that SERQ is robust to the calibration dataset size and dataset characteristics, achieving similar perplexity scores across all settings.

**Evaluation on Generation Tasks.** While prior work on linear layer quantization primarily focuses on prefill-sensitive tasks (See section 4.1), we additionally evaluate generation tasks to demonstrate

Table 6: Generation task evaluation with GSM8K and LongBench datasets.

| Model | #Bits | Method | GSM8K 5-shot(↑) | | LongBench(↑) | | |
| | | | flexible extract | strict match | qmsum | samsum | repobench-p |
|---|---|---|---|---|---|---|---|
| LLaMA-3 8B | FP16 | baseline | 48.07 | 47.69 | 23.43 | 46.26 | 66.56 |
| | W4A8 | LLM.int4() | 7.88 | 4.62 | 13.29 | 34.2 | 44.93 |
| | | L$^2$QER | 36.24 | 35.33 | 7.28 | 25.81 | 47.03 |
| | | SERQ | **42.61** | **42.38** | **22.97** | **44.27** | **63.34** |
| | W4A4 | L$^2$QER | 7.96 | 7.66 | 0.09 | 1.98 | 5.65 |
| | | SERQ | **23.65** | **23.12** | **19.08** | **40.85** | **54.78** |
| LLaMA-3.2 3B | FP16 | baseline | 26.08 | 25.93 | 23.94 | 42.98 | 64.42 |
| | W4A8 | LLM.int4() | 6.76 | 6.44 | 15.83 | 36.42 | 35.24 |
| | | L$^2$QER | 18.95 | 18.42 | 21.53 | 40.63 | 60.18 |
| | | SERQ | **21.68** | **21.23** | **22.25** | **42.47** | **60.78** |
| | W4A4 | L$^2$QER | 3.56 | 2.73 | 14.92 | 33.87 | 41.99 |
| | | SERQ | **16.22** | **15.77** | **19.31** | **42.14** | **53.67** |

Table 7: Effect of static activation flattening (SAF) on perplexity.

| #Bits | #Q-config. | Method | LLaMA-2 7B | LLaMA-3.2 1B | LLaMA-3.2 3B | Qwen-2.5 3B |
|---|---|---|---|---|---|---|
| FP16 | | baseline | 5.47 | 9.75 | 7.81 | 8.03 |
| W4A8 | INT (RTN) | only-SAF | 6.64 | 25.49 | 12.75 | 12.41 |
| | | SERQ w/o SAF | 5.64 | 11.13 | 8.37 | **8.56** |
| | | SERQ | **5.63** | **11.1** | **8.36** | 8.57 |
| W4A4 | INT (RTN) | only-SAF | 7.58 | 56.92 | 18.84 | 20.67 |
| | | SERQ w/o SAF | 6.05 | 13.6 | **9.36** | 10.83 |
| | | SERQ | **6.03** | **13.49** | 9.42 | **9.57** |
| | MXFP4 | only-SAF | 7.03 | 17.21 | 11.49 | 12.88 |
| | | SERQ w/o SAF | **6.21** | 13.56 | **9.59** | 10.4 |
| | | SERQ | 6.22 | **13.53** | 9.69 | **9.84** |

accuracy in generation. We conduct experiments on GSM8K (Cobbe et al. (2021)) and LongBench (Bai et al. (2024)) under W4A8 and W4A4 settings using LLaMA-3 8B and LLaMA-3.1 3B models. Compared with other matrix decomposition methods, SERQ achieves superior performance with only minor accuracy degradation in the W4A8 setting. Although the W4A4 setting leads to a notable accuracy drop overall, SERQ maintains reliable results where other methods fail.

**Ablation Study on Static Activation Flattening.** In SERQ, static activation flattening (SAF) is combined with saliency aware low rank error reconstruction, effectively stabilizing the activation distribution during quantization. To isolate the contribution of each component, we conduct an ablation study, as summarized in Table 7. For the LLaMA-2-7B model, SERQ without SAF maintains accuracy well when only low error reconstruction is applied. However, for more error-sensitive small-scale models such as Qwen-2.5-3B, SAF provides a substantial accuracy improvement. This behavior arises because smaller models accumulate fewer partial sums per output element, making quantization errors from a single group disproportionately influential on the final output. Results from the MXFP4 implementation exhibit a similar trend. Although the smaller grouping in MXFP4 partially mitigates outlier effects, SAF still yields additional accuracy gains, demonstrating the importance of this technique and the potential of SERQ for edge-oriented LLM deployment.

## 5 CONCLUSION

In this work, we introduced SERQ, a saliency-aware error reconstruction method that enables end-to-end 4-bit quantization of both weights and activations using a single low rank matrix. By combining static activation flattening with mergeable weight permutation, SERQ identifies salient rows in the weight matrix without incurring additional latency overhead and reconstructs them through an auxiliary low rank branch. SERQ consistently outperforms prior matrix decomposition and distribution flattening methods, including state-of-the-art LoRA-based and rotation-based approaches, in terms of accuracy. Our implementation with the MXFP4 data format on NVIDIA Blackwell architecture further demonstrates that the auxiliary branch introduces minimal latency overhead, achieving significant speedups while preserving accuracy in low precision computation.

ACKNOWLEDGMENTS

This work was partly supported by the National Research Foundation of Korea (NRF) grant funded by the Korea government (MSIT) (RS-2024-00352468), K-CHIPS (Korea Collaborative & High-tech Initiative for Prospective Semiconductor Research) (2410000620, RS-2024-00405946, 24052-15TC) funded by the Ministry of Trade, Industry & Energy (MOTIE, Korea), and Institute of Information & communications Technology Planning & Evaluation (IITP) grant funded by the Korea government(MSIT) (RS-2025-02263669, Development of server-level DRAM-stacked PIM solution for accelerating ultra-large AI models).

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

## A  Appendix

### A.1  Experiments on Low-Rank Saliency

In this analysis, we vary the number of weight rows included in SVD, selected by saliency in descending order of activation channel scales, to examine the trade-off between loss from rank reduction and the coverage of error reconstruction. To isolate the quantization effects of the weight matrix, we apply W4A16 quantization with a group size of 128 and fix the SVD rank size to 64. Figure 4 shows that decomposing only salient weights is more effective for reducing quantization error, as the perplexity score decreases slightly on the leftmost side when using row size 64. This result indicates that restricting reconstruction to salient rows not only avoids accuracy degradation but also yields consistent improvement across all tested LLaMA models. We observe about 1 to 4 percent improvements compared to using the same limited rank to cover the full weight matrix. Expanding beyond the salient subset offers little or even negative benefit, since the additional low rank factorization loss outweighs the marginal coverage. These findings motivate a single-layer reconstruction, rather than two low-rank factors, and support a saliency-aware error reconstruction strategy that keeps both the main and auxiliary paths in pure 4-bit, achieving efficiency without sacrificing accuracy.

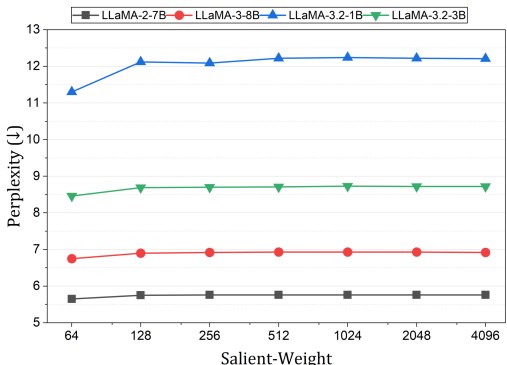

Figure 4: The trade-off between loss from rank reduction and the coverage of error reconstruction. The figure shows that higher accuracy is achieved by reconstructing errors for salient rows with smaller ranks, rather than covering a larger portion of the weight matrix.

### A.2  Related Works

#### A.2.1  Low-Rank Adaptor for Quantization Error Reconstruction

Low-Rank Adaptation (LoRA)(Hu et al. (2021)), originally introduced as a representative Parameter-Efficient Fine-Tuning (PEFT) technique, has demonstrated remarkable effectiveness in reducing training overhead while facilitating rapid adaptation to down-stream tasks. Its architecture, which injects two trainable low-rank matrices into the pre-trained weights, exhibits high compatibility with existing transformer models. Leveraging this property, recent LLM quantization utilizes these auxiliary low-rank matrices to reconstruct and compensate for quantization errors inherent in compressed weights. In particular, the low-rank structure of LoRA align intrinsically with Singular Value Decomposition (SVD), which yields the optimal solution for low-rank approximation. The approximation of the quantization error, $\boldsymbol{E} = \boldsymbol{W} - \mathrm{Q}(\boldsymbol{W})$, can be formally defined using truncated SVD as follows:

$$\boldsymbol{E} \approx \boldsymbol{U}_r \Sigma_r \boldsymbol{V}^T = \boldsymbol{L}_1 \boldsymbol{L}_2 \tag{8}$$

where usually $\boldsymbol{U}_r \Sigma_r^{\frac{1}{2}}$ and $\Sigma_r^{\frac{1}{2}} \boldsymbol{V}_r^T$ correspond to the low-rank factors $\boldsymbol{L}_1$ and $\boldsymbol{L}_2$, respectively, and $r$ denotes the rank. This paradigm of LoRA-based quantization has been extensively explored in both weight-only and weight-activation quantization regimes.

**Weight-only Quantization.** ZeroQuant-V2(Yao et al. (2023)) and LQER(Zhang et al. (2024)) demonstrated that integrating optimized low-rank matrices derived via SVD with quantized weights effectively mitigates quantization errors. This methodologies was further advanced by CALDERA(Saha et al. (2024)) and QERA(Zhang et al. (2025)). CALDERA(Saha et al. (2024))

initializes low-rank matrices using SVD and employs an iterative update scheme for both the quantization error and the low-rank adaptors, thereby maintaining competitive performance even under ultra-low quantization(e.g., ~2-bits). QERA(Zhang et al. (2025)) addresses the limitations of heuristic approach to incorporating activation sensitivity by deriving a closed-form solution that effectively minimizes the quantization error of the linear layer's output.

**Weight-Activation Quantization.** A critical challenge in weight-activation quantization is how to manage outliers present in activation channels. $L^2$QER(Zhang et al. (2024)) and ASER(Zhao et al. (2024)) propose mixed-precision configurations (e.g., W4A8 or W4A6) to address this issue. To specifically mitigate activation outliers, $L^2$QER(Zhang et al. (2024)) applies per-channel activation scaling, while absorbing the corresponding scaling factors into the SVD-based low-rank matrices. Similarly, ASER(Zhao et al. (2024)) enhances the efficiency of low-rank approximation via whitening SVD and alleviates outlier effects through activation smoothing. Furthermore, SVDQuant(Li et al. (2025)) migrates the quantization difficulty of activations to the low-rank adaptor while establishing a hardware-efficient pipeline, demonstrating superior quantization framework particularly in diffusion models().

### A.2.2 MICROSCALING (MX) FORMAT

The Microscaling (MX) format Rouhani et al. (2023) is a block-scaled, low-precision representation that associates each small block with a single scale and quantized elements, thereby retaining dynamic range while enabling efficient 4–8 bit computation. Concretely, data are partitioned into blocks; each block stores one scale, estimated from its maximum value, and the elements within the block are stored in low-bit precision—either in floating-point formats (MXFP8, MXFP6, MXFP4) or in integer format (MXINT8). Note that INT4 is not included among MX variants. All elements are defined relative to the block's shared scale, which is fixed at 8-bit precision.

In contrast, naive single-scale low-bit formats often degrade sharply on LLMs: a few outliers inflate the global scale, wasting quantization levels and clipping inlier values. By adopting a block-level granularity (typically set to 32), MX preserves usable dynamic range and maintains effective quantization. As a result, MX-based models achieve low-precision inference with minimal accuracy loss, reduced memory and bandwidth, and hardware-friendly execution, since computations reduce to standard low-bit arithmetic plus lightweight per-block scaling. Owing to its effectiveness for low-precision quantization in deep learning, NVIDIA's Blackwell architecture, equipped with fifth-generation Tensor Cores, provides CUTLASS kernels supporting MX data formats for LLM workloads.

### A.3 GPTQ IMPLEMENTATION

Recently, GPTQ(Frantar et al. (2023)) has demonstrated strong performance in weight-only quantization through a lightweight, Hessian-guided optimization. Specifically, GPTQ(Frantar et al. (2023)) quantize weights sequentially while utilizing Hessian information to compensate for the induced quantization error by remaining (unquantized) weights. The weights update is defined as follows:

$$w_j \leftarrow w_j - e_j \cdot \frac{[\boldsymbol{H}^{-1}]_{ji}}{[\boldsymbol{H}^{-1}]_{ii}} \tag{9}$$

Where $i$ denotes the index of the weight currently being quantized, and $j$ represents the indices of the remaining unquantized weights (i.e., $j > i$). Here, $e_i = w_i - Q(w_i)$ corresponds to the quantization error induced at the current step, and $\boldsymbol{H}^{-1}$ refers to the inverse Hessian matrix, which guides the optimal compensation for the quantization error. However, this update process is not orthogonal to the low-rank error reconstruction targeting $\boldsymbol{E} = \boldsymbol{W} - Q(\boldsymbol{W})$. This is because the iterative updates in GPTQ shift the weight distribution, causing it to diverge from the original statistics. To address this, we modify the error reconstruction pipeline to achieve a GPTQ-friendly decomposition. we prioritize extracting the high-variance components from the salient rows into a 4-bit low-rank matrix prior to the GPTQ process. Specifically, we define the low-rank matrix $\boldsymbol{R}$ as $Q(\widetilde{\boldsymbol{W}_s})$ and replace the original salient rows with the residual error $\widetilde{\boldsymbol{W}_s} - \boldsymbol{R}$. This substitution effectively migrates the flattening-driven outlier away from the main weights, thereby stabilizing the weight distribution. Furthermore, it reduces computational complexity by eliminating the need for dequantization-requantization process during error reconstruction. In this way, GPTQ(Frantar et al.

Table 8: Comparison of Quantization Signal-to-Noise Ratio (QSNR) between SERQ and SVD.

| Module | Rank | Method | QSNR($\uparrow$) | Module | Rank | Method | QSNR($\uparrow$) |
|---|---|---|---|---|---|---|---|
| LLaMA-3 8B $10^{th}$ o_proj | 128 | SVD | 19.87 | LLaMA-3 8B $20^{th}$ o_proj | 128 | SVD | 18.91 |
| | | SERQ w/o SAF | 25.91 | | | SERQ w/o SAF | 21.24 |
| | 64 | SVD | 19.41 | | 64 | SVD | 18.7 |
| | | SERQ w/o SAF | 25.49 | | | SERQ w/o SAF | 20.81 |
| | 32 | SVD | 19.21 | | 32 | SVD | 18.58 |
| | | SERQ w/o SAF | 20.22 | | | SERQ w/o SAF | 18.46 |
| | 16 | SVD | 19.06 | | 16 | SVD | 18.52 |
| | | SERQ w/o SAF | 20.42 | | | SERQ w/o SAF | 18.54 |
| | 8 | SVD | 19.01 | | 8 | SVD | 18.49 |
| | | SERQ w/o SAF | 21.29 | | | SERQ w/o SAF | 18.75 |

(2023)) targets a more quantization-amenable main weight matrix, while the low-rank path captures the salient components.

### A.4 ANALYSIS OF QUANTIZATION ERROR RECONSTRUCTION.

Quantization Error Reconstruction aims to approximate and compensate for the quantization error of the main weights by leveraging the LoRA architecture. A pivotal aspect of this process is the selection of the saliency metric used to minimize the quantization error. Truncated SVD approaches approximate the residual between the original weights and their quantized counterparts, optimizing the following objective function:

$$\min_{\boldsymbol{C}} \|\boldsymbol{E} - \boldsymbol{C}\|_F^2 = \sum_{i,j}(\boldsymbol{E}_{ij} - \boldsymbol{C}_{ij})^2 \tag{10}$$

where $\boldsymbol{E}$ represents the quantization error and $\boldsymbol{C}$ is the low-rank compensate term. To minimize this objective, truncated SVD employs singular values as the saliency metric, providing a closed-form solution via truncated SVD. However, in Transformer models, the outputs of linear operations (i.e., $\boldsymbol{Y} = \boldsymbol{XW}$) propagate sequentially through the layers. Consequently, the cumulative quantization error that directly impacts model performance is effectively defined by the output error, i.e., $\Delta\boldsymbol{Y} = \boldsymbol{X}\Delta\boldsymbol{W}$. Therefore, while minimizing the weight error $\Delta\boldsymbol{W}$ is necessary, it is far more critical to mitigate the error contribution from indices $i$ where the activation inputs $\boldsymbol{X}$ possess large magnitudes. To validate this theoretical justification, we present a comparison of the Quantization Signal-to-Noise Ratio (QSNR) for the linear layer outputs in Table 8. We specifically compare the truncated SVD method against our SERQ (w/o SAF) under a weight-only quantization setting. The results demonstrate that SERQ (w/o SAF) yields superior quantization performance compared to SVD at identical rank budgets. Notably, we observe a sharp increase in QSNR for SERQ (w/o SAF) beyond a specific rank threshold. This indicates that reducing the rank budget below this critical point fails to capture all salient weights. Consequently, to ensure the stability of model performance, we selected a rank of 128.

### A.5 ADDITIONAL COMPARISON WITH SOTA ROTATION METHODS.

Although our primary research focus lies in LoRA-based quantization, the proposed SERQ method marks a significant breakthrough by enabling a robust W4A4 quantization scheme. Given that we quantize both activations and weights to 4-bit, we provide a comprehensive comparison against state-of-the-art (SOTA) rotation-based methods, which currently dominate the W4A4 quantization. In Table 9, we present performance benchmarks alongside calibration costs for leading rotation-based techniques, including SpinQuant(Liu et al. (2025)), FlatQuant(Sun et al. (2025)), DuQuant(Lin et al. (2024a)), OSTQuant(Hu et al. (2025)), and QuaRot(Ashkboos et al. (2024)). The results indicate that methods employing extensive training or heavy iterative optimization generally yield the highest accuracy. FlatQuant(Sun et al. (2025)), in particular, demonstrates superior performance, outperforming all the compared models. However, it is important to note that this advantage comes at the expense of optimizing a large number of trainable parameters and introducing floating-point (FP) affine transformations during runtime, which can incur additional computational overhead. Regarding calibration efficiency, the training-free approaches—specifically SERQ and QuaRot(Ashkboos

Table 9: Comparison of SERQ with SOTA rotation-based quantization methods in terms of calibration time and model accuracy (PPL, 0-shot, MMLU).

| Method | LLaMA-2 7B | | | | LLaMA-2 13B | | | | LLaMA-3.2 3B | | | |
|---|---|---|---|---|---|---|---|---|---|---|---|---|
| | Calib. Time | PPL ($\downarrow$) | 0-shot ($\uparrow$) | MMLU ($\uparrow$) | Calib. Time | PPL ($\downarrow$) | 0-shot ($\uparrow$) | MMLU ($\uparrow$) | Calib. Time | PPL ($\downarrow$) | 0-shot ($\uparrow$) | MMLU ($\uparrow$) |
| FP16 | | 5.47 | 64.09 | 41.83 | | 4.88 | 66.53 | 52.04 | | 7.81 | 62.66 | 54.06 |
| SpinQuant | ∼598m | 6 | 61 | 34.8 | ∼721m | 5.2 | 64.8 | 47.8 | ∼156m | 10.15 | 56.88 | 42.42 |
| FlatQuant | ∼131m | 5.75 | 62.11 | 38.24 | ∼225m | 5.08 | 64.94 | 49.59 | ∼76m | 8.51 | 60.41 | 50 |
| DuQuant | ∼255m | 5.95 | 61.84 | 33.46 | ∼486m | 5.23 | 64.88 | 48.59 | ∼140m | 9.56 | 58.31 | 44.79 |
| OstQuant | ∼72m | 5.91 | 61.72 | 37.39 | ∼92m | 5.26 | 64.92 | 47.77 | ∼26m | 9.04 | 58.81 | 46.05 |
| QuaRot | ∼31m | 6.15 | 59.53 | 33.58 | ∼44m | 5.41 | 62.55 | 47.25 | ∼8m | 9.73 | 55.76 | 44.75 |
| SERQ | ∼23m | 5.97 | 61.87 | 37.03 | ∼48m | 5.2 | 64.82 | 47.17 | ∼15m | 9.15 | 58.5 | 45.7 |

et al. (2024))—offer drastically lower calibration costs. What is particularly compelling is that, despite this negligible overhead, SERQ delivers performance that fully comparable to computationally intensive, training-based methods. This confirms that SERQ acts as a highly efficient, lightweight framework, offering an optimal trade-off between calibration cost and model accuracy.

## A.6 GPU PERFORMANCE ANALYSIS DETAILS

This section reports the absolute latency obtained with NVIDIA Blackwell CUTLASS kernels when operating on linear layers of different sizes used in LLM models. We also provide the inference overhead introduced by low rank factors, where SERQ includes an additional multiplication path using $R$, and L$^2$QER includes the factors $L_1$ and $L_2$.

## A.7 OVERALL LLM ACCURACY RESULTS

We report overall accuracy results on the evaluated benchmarks, which are not captured in Tables 1 and 2 since they only present average values. Tables 7 and 8 provide detailed accuracy results for low-rank matrix decomposition methods under the W4A8 and W4A4 settings, respectively. Table 9 reports detailed results for different distribution flattening methods.

## A.8 LLM USAGE

In this paper, we used a large language model to aid and polish the manuscript to improve the overall writing quality.

Table 10: GPU latency results in linear layers.

| Sequence Length | Weight Size | FP16($\mu$s) | MXFP4($\mu$s) | Inference overhead (MXFP4) | | | |
| | | | | $R(\mu$s) | $L_1(\mu$s) | $L_2(\mu$s) | $L_1 L_2(\mu$s) |
|---|---|---|---|---|---|---|---|
| 2048 | $2048 \times 2048$ | 71.402 | 27.457 | 10.501 | 16.606 | 12.009 | 28.615 |
| | $2048 \times 8192$ | 207.482 | 72.7 | 20.934 | 16.606 | 24.86 | 41.466 |
| | $8192 \times 2048$ | 212.87 | 88.738 | 10.501 | 43.235 | 12.009 | 55.244 |
| | $4096 \times 4096$ | 211.894 | 70.255 | 14.637 | 26.833 | 14.66 | 41.493 |
| | $4096 \times 11008$ | 519.322 | 181.359 | 27.0112 | 26.833 | 31.034 | 57.867 |
| | $11008 \times 4096$ | 512.25 | 175.964 | 14.637 | 55.51 | 14.66 | 70.171 |
| | $4096 \times 14336$ | 643.27 | 235.397 | 38.342 | 26.833 | 40.638 | 67.472 |
| | $14336 \times 4096$ | 655.757 | 226.379 | 14.637 | 69.887 | 14.659 | 84.547 |
| | $8192 \times 8192$ | 755.494 | 260.054 | 20.934 | 43.235 | 24.86 | 68.095 |
| | $8192 \times 28672$ | 2604.973 | 946.403 | 174.537 | 43.235 | 174.857 | 218.09 |
| | $28672 \times 8192$ | 2444.912 | 946.787 | 20.934 | 132.005 | 24.86 | 156.865 |
| 4096 | $2048 \times 2048$ | 122.298 | 37.664 | 14.4035 | 16.669 | 14.662 | 31.331 |
| | $2048 \times 8192$ | 403.859 | 142.159 | 56.211 | 16.669 | 55.991 | 72.66 |
| | $8192 \times 2048$ | 394.138 | 132.903 | 14.4035 | 43.292 | 14.662 | 57.954 |
| | $4096 \times 4096$ | 382.79 | 136.076 | 20.936 | 26.805 | 25.155 | 51.96 |
| | $4096 \times 11008$ | 1009.898 | 355.986 | 129.704 | 26.8045 | 129.876 | 156.681 |
| | $11008 \times 4096$ | 986.624 | 345.739 | 20.938 | 55.947 | 25.155 | 81.1 |
| | $4096 \times 14336$ | 1262.093 | 477.976 | 175.101 | 26.8045 | 175.268 | 202.073 |
| | $14336 \times 4096$ | 986.97 | 445.022 | 20.938 | 70.016 | 25.155 | 95.171 |
| | $8192 \times 8192$ | 1560.854 | 506.364 | 56.211 | 43.292 | 55.991 | 99.283 |
| | $8192 \times 28672$ | 4868.259 | 1911.05 | 355.274 | 43.292 | 355.969 | 399.261 |
| | $28672 \times 8192$ | 4849.363 | 1853.27 | 56.211 | 132.019 | 55.991 | 188.01 |

Table 11: Accuracy results of low-rank matrix decomposition methods when tested with W4A8 settings.

| Model | Methods | PPL (↓) | 0-Shot Common Sense Reasoning tasks | | | | | | | | | MMLU | | | | |
|---|---|---|---|---|---|---|---|---|---|---|---|---|---|---|---|---|
| | | | BoolQ (↑) | PIQA (↑) | SIQA (↑) | ARC-e (↑) | ARC-c (↑) | HellaS. (↑) | WinoG. (↑) | OBQA (↑) | Avg. (↑) | Human. (↑) | Other (↑) | SocialS. (↑) | STEM (↑) | Avg. (↑) |
| LLaMA-2-7B | baseline | 5.47 | 77.74 | 79.05 | 46.11 | 74.49 | 46.25 | 76 | 68.9 | 44.2 | 64.09 | 39.72 | 47.12 | 47.45 | 34.28 | 41.83 |
| | LLM.int4() | 7.28 | 69.3 | 76.39 | 43.76 | 68.48 | 43.43 | 71.19 | 64.96 | 39.6 | 59.64 | 26.23 | 27.9 | 33.54 | 29.84 | 29.01 |
| | L²QER | 5.83 | 75.99 | 78.51 | 44.98 | 74.75 | 44.03 | 75.14 | 68.98 | 44.4 | 63.35 | 37.39 | 43.13 | 43.48 | 34.76 | 39.4 |
| | SERQ (RTN) | 5.64 | 76.85 | 78.4 | 44.93 | 73.7 | 45.31 | 75.12 | 68.75 | 44.2 | 63.41 | 37.39 | 45.93 | 45.47 | 33.65 | 40.21 |
| | SERQ (GPTQ) | 5.59 | 77 | 78.78 | 44.83 | 73.15 | 44.2 | 75.05 | 68.27 | 43 | 63.04 | 37.9 | 45.22 | 44.85 | 34.54 | 40.29 |
| LLaMA-2-13B | baseline | 4.88 | 80.61 | 80.52 | 47.39 | 77.53 | 49.23 | 79.38 | 72.38 | 45.2 | 66.53 | 47.89 | 59.29 | 61.16 | 42.21 | 52.04 |
| | LLM.int4() | 5.35 | 79.91 | 79.38 | 46.52 | 75.34 | 49.15 | 77.46 | 69.69 | 42.6 | 65.01 | 44.31 | 53.07 | 54.76 | 38 | 47.12 |
| | L²QER | 5.08 | 81.41 | 79.87 | 46.37 | 76.85 | 50.17 | 78.52 | 72.3 | 45.6 | 66.39 | 46.63 | 58.71 | 59.02 | 42.02 | 50.98 |
| | SERQ (RTN) | 5 | 80.18 | 80.36 | 46.47 | 76.3 | 48.38 | 78.7 | 71.67 | 44.8 | 65.86 | 47.08 | 57.64 | 58.99 | 42.09 | 50.9 |
| | SERQ (GPTQ) | 4.98 | 79.54 | 79.38 | 46.26 | 76.18 | 47.27 | 78.83 | 72.45 | 45.4 | 65.66 | 45.59 | 55.52 | 58.01 | 40.91 | 49.46 |
| LLaMA-3-8B | baseline | 6.13 | 81.07 | 80.74 | 47.08 | 77.69 | 52.99 | 79.2 | 73.48 | 45 | 67.16 | 54.81 | 70.55 | 73.25 | 53.89 | 62.13 |
| | LLM.int4() | 10.82 | 75.23 | 74.97 | 44.22 | 67.38 | 42.24 | 72.63 | 70.01 | 40.2 | 60.86 | 42.34 | 53.91 | 52.65 | 40.41 | 46.72 |
| | L²QER | 7.16 | 80.67 | 79.54 | 45.29 | 75.34 | 51.28 | 77.01 | 72.53 | 43.8 | 65.68 | 52.01 | 66.4 | 67.66 | 48.37 | 57.81 |
| | SERQ (RTN) | 6.71 | 81.41 | 80.2 | 45.7 | 77.44 | 50.77 | 78.25 | 73.64 | 44.4 | 66.4 | 51.9 | 67.94 | 69.16 | 48.59 | 58.49 |
| | SERQ (GPTQ) | 6.52 | 80.8 | 80.9 | 46.21 | 76.52 | 51.02 | 78.25 | 72.53 | 43.6 | 66.23 | 53.07 | 69.46 | 70.62 | 51.76 | 60.25 |
| LLaMA-3.2-1B | baseline | 9.75 | 63.67 | 74.48 | 42.84 | 60.44 | 36.18 | 63.73 | 59.83 | 37.4 | 54.82 | 34.92 | 41.1 | 39.78 | 32.29 | 36.76 |
| | LLM.int4() | 32.39 | 51.07 | 63.82 | 38.74 | 42.09 | 28.24 | 46.89 | 55.25 | 30.4 | 44.56 | 25.1 | 24.24 | 23.17 | 22.96 | 24.01 |
| | L²QER | 12.06 | 60.67 | 70.73 | 41.45 | 53.45 | 32.68 | 59.04 | 56.91 | 32.8 | 50.97 | 28.59 | 35.92 | 33.99 | 27.85 | 31.23 |
| | SERQ (RTN) | 11.14 | 62.75 | 72.85 | 42.07 | 56.69 | 34.39 | 60.74 | 56.35 | 36.2 | 52.76 | 32.56 | 36.53 | 33.44 | 28.7 | 32.77 |
| | SERQ (GPTQ) | 10.45 | 63.24 | 72.47 | 42.02 | 58.92 | 36.95 | 61.7 | 58.96 | 37.2 | 53.93 | 27.72 | 29 | 27.72 | 23.69 | 27.1 |
| LLaMA-3.2-3B | baseline | 7.81 | 72.97 | 77.53 | 46.93 | 71.59 | 46.25 | 73.49 | 69.53 | 43 | 62.66 | 48.78 | 63.08 | 62.59 | 44.72 | 54.06 |
| | LLM.int4() | 12.47 | 67.55 | 73.99 | 42.94 | 62.58 | 37.46 | 66.57 | 61.8 | 35.6 | 56.06 | 37.26 | 46.6 | 47.71 | 36.63 | 41.48 |
| | L²QER | 8.78 | 71.8 | 76.61 | 45.8 | 66.29 | 41.47 | 71.13 | 67.8 | 42 | 60.36 | 46.93 | 58.32 | 59.67 | 42.47 | 51.24 |
| | SERQ (RTN) | 8.38 | 71.22 | 76.55 | 46.16 | 69.19 | 44.28 | 72.31 | 69.85 | 40.6 | 61.27 | 46.91 | 59.86 | 59.02 | 42.53 | 51.45 |
| | SERQ (GPTQ) | 8.18 | 71.19 | 76.44 | 46.83 | 68.9 | 45.39 | 72.4 | 69.14 | 42 | 61.54 | 46.76 | 58.93 | 56.65 | 40.63 | 50.24 |
| Qwen-2.5 3B | baseline | 8.03 | 77.58 | 78.73 | 50.1 | 73.11 | 47.1 | 73.62 | 68.51 | 41.8 | 63.82 | 56.77 | 71.07 | 76.15 | 60.96 | 65.12 |
| | LLM.int4() | 9.98e+5 | 37.86 | 51.52 | 34.19 | 24.58 | 26.19 | 26.51 | 51.3 | 32.6 | 35.59 | 27.14 | 24.01 | 23.79 | 26.26 | 25.52 |
| | L²QER | 12.25 | 68.1 | 73.07 | 42.99 | 47.22 | 37.54 | 66.57 | 64.01 | 38.2 | 54.71 | 26.82 | 26.97 | 25.22 | 24.58 | 26 |
| | SERQ (RTN) | 8.67 | 76.85 | 77.75 | 50.41 | 76.68 | 48.38 | 72.13 | 65.82 | 42.2 | 63.78 | 55.39 | 68.46 | 73.71 | 57.6 | 62.79 |
| | SERQ (GPTQ) | 8.36 | 77.68 | 77.86 | 49.23 | 75.46 | 48.63 | 72.35 | 66.61 | 41.8 | 63.7 | 55.07 | 69.46 | 74.65 | 58.2 | 63.25 |

Table 12: Accuracy results of low-rank matrix decomposition methods when tested with W4A4 settings.

**LLaMA-2-7B**

| Model | Methods | PPL (↓) | BoolQ (↑) | PIQA (↑) | SIQA (↑) | ARC-e (↑) | ARC-c (↑) | HellaS. (↑) | WinoG. (↑) | OBQA (↑) | Avg. (↑) | Human. (↑) | Other (↑) | SocialS. (↑) | STEM (↑) | Avg. (↑) |
|---|---|---|---|---|---|---|---|---|---|---|---|---|---|---|---|---|
| LLaMA-2-7B | baseline | 5.47 | 77.74 | 79.05 | 46.11 | 74.49 | 46.25 | 76 | 68.9 | 44.2 | 64.09 | 39.72 | 47.12 | 47.45 | 34.28 | 41.83 |
| | LLM.int4() | 6.32e+2 | 43.3 | 49.84 | 34.08 | 27.44 | 25.43 | 26.46 | 50.28 | 22 | 34.85 | 25.04 | 23.46 | 24.6 | 24.2 | 24.41 |
| | L²QER | 7.37 | 66.73 | 76.28 | 43.09 | 65.95 | 39.51 | 70.24 | 61.56 | 38 | 57.67 | 33.41 | 39.59 | 38.48 | 30.45 | 29.63 |
| | L²QER-MXFP4 | 6.3 | 73.33 | 76.71 | 44.37 | 70.71 | 42.41 | 72.3 | 66.93 | 40.8 | 60.95 | 33.41 | 39.59 | 38.48 | 30.45 | 35.22 |
| | SERQ (RTN) | 6.03 | 75.96 | 77.26 | 43.96 | 72.26 | 44.8 | 73.71 | 65.98 | 40.2 | 61.77 | 35.28 | 42.87 | 41.7 | 33.78 | 38.03 |
| | SERQ (GPTQ) | 5.97 | 74.56 | 78.13 | 44.22 | 72.9 | 43.09 | 73.88 | 66.61 | 41.6 | 61.87 | 34.67 | 41.33 | 40.92 | 32.54 | 37.03 |
| | SERQ-MXFP4 | 6.22 | 73 | 77.2 | 44.17 | 70.2 | 43.69 | 72.59 | 68.19 | 41 | 61.26 | 32.16 | 40.01 | 39.1 | 31.43 | 35.25 |

**LLaMA-2-13B**

| Model | Methods | PPL (↓) | BoolQ (↑) | PIQA (↑) | SIQA (↑) | ARC-e (↑) | ARC-c (↑) | HellaS. (↑) | WinoG. (↑) | OBQA (↑) | Avg. (↑) | Human. (↑) | Other (↑) | SocialS. (↑) | STEM (↑) | Avg. (↑) |
|---|---|---|---|---|---|---|---|---|---|---|---|---|---|---|---|---|
| LLaMA-2-13B | baseline | 4.88 | 80.61 | 80.52 | 47.39 | 77.53 | 49.23 | 79.38 | 72.38 | 45.2 | 66.53 | 47.89 | 59.29 | 61.16 | 42.21 | 52.04 |
| | LLM.int4() | 2.21e+3 | 40.24 | 49.89 | 35.11 | 26.64 | 26.19 | 26.08 | 49.8 | 24 | 34.74 | 24.1 | 23.04 | 22.26 | 22.2 | 23.04 |
| | L²QER | 6.27 | 74.31 | 76.99 | 44.42 | 71.25 | 44.88 | 73.71 | 63.61 | 40 | 61.15 | 38.55 | 44.06 | 45.73 | 35.68 | 40.7 |
| | L²QER-MXFP4 | 5.46 | 77.86 | 78.13 | 46.47 | 74.37 | 47.1 | 75.8 | 70.24 | 43 | 64.12 | 43.34 | 52.98 | 54.6 | 40.37 | 47.27 |
| | SERQ (RTN) | 5.24 | 78.35 | 80.25 | 46.37 | 76.39 | 46.42 | 78.11 | 70.64 | 44.6 | 65.14 | 45.89 | 55.29 | 56.45 | 41.14 | 49.22 |
| | SERQ (GPTQ) | 5.2 | 78.29 | 29.27 | 46.06 | 75.63 | 46.5 | 78.01 | 70.01 | 44.8 | 64.82 | 43.83 | 53.3 | 55.48 | 38 | 47.17 |
| | SERQ-MXFP4 | 5.39 | 77.86 | 78.73 | 45.19 | 75.59 | 48.12 | 76.18 | 70.09 | 43 | 64.35 | 43.59 | 52.33 | 54.6 | 40.28 | 47.19 |

**LLaMA-3-8B**

| Model | Methods | PPL (↓) | BoolQ (↑) | PIQA (↑) | SIQA (↑) | ARC-e (↑) | ARC-c (↑) | HellaS. (↑) | WinoG. (↑) | OBQA (↑) | Avg. (↑) | Human. (↑) | Other (↑) | SocialS. (↑) | STEM (↑) | Avg. (↑) |
|---|---|---|---|---|---|---|---|---|---|---|---|---|---|---|---|---|
| LLaMA-3-8B | baseline | 6.13 | 81.07 | 80.74 | 47.08 | 77.69 | 52.99 | 79.2 | 73.48 | 45 | 67.16 | 54.81 | 70.55 | 73.25 | 53.89 | 62.13 |
| | LLM.int4() | 4.87e+2 | 50.89 | 50.22 | 32.65 | 27.95 | 23.29 | 28.29 | 48.3 | 27.4 | 36.12 | 25.16 | 23.56 | 22.55 | 22.39 | 23.61 |
| | L²QER | 11.44 | 61.71 | 72.63 | 42.02 | 62.37 | 39.59 | 67.29 | 60.54 | 37.4 | 55.44 | 34.06 | | | | 38.33 |
| | L²QER-MXFP4 | 7.83 | 76.12 | 77.97 | 44.01 | 74.66 | 48.29 | 74.6 | 69.38 | 41.6 | 63.33 | 48.57 | 61.15 | 62.89 | 45.58 | 53.82 |
| | SERQ (RTN) | 8.07 | 74.5 | 76.06 | 44.37 | 71.93 | 46.67 | 72.98 | 70.64 | 42.8 | 62.49 | 46.14 | 60.22 | 60.32 | 43.83 | 51.84 |
| | SERQ (GPTQ) | 7.75 | 76.85 | 77.8 | 43.55 | 72.43 | 44.88 | 73.6 | 69.38 | 40.8 | 62.41 | 48.42 | 61.6 | 62.2 | 45.92 | 53.8 |
| | SERQ-MXFP4 | 7.63 | 76.15 | 77.2 | 44.11 | 72.69 | 45.39 | 75.13 | 68.43 | 42.6 | 62.71 | 47.72 | 60.7 | 63.18 | 46.84 | 53.48 |

**LLaMA-3.2-1B**

| Model | Methods | PPL (↓) | BoolQ (↑) | PIQA (↑) | SIQA (↑) | ARC-e (↑) | ARC-c (↑) | HellaS. (↑) | WinoG. (↑) | OBQA (↑) | Avg. (↑) | Human. (↑) | Other (↑) | SocialS. (↑) | STEM (↑) | Avg. (↑) |
|---|---|---|---|---|---|---|---|---|---|---|---|---|---|---|---|---|
| LLaMA-3.2-1B | baseline | 9.75 | 63.67 | 74.48 | 42.84 | 60.44 | 36.18 | 63.73 | 59.83 | 37.4 | 54.82 | 34.92 | 41.1 | 39.78 | 32.29 | 36.76 |
| | LLM.int4() | 1.7e+3 | 41.13 | 51.03 | 33.73 | 26.89 | 22.61 | 26.92 | 51.38 | 27.8 | 35.19 | 24.21 | 24.59 | 24.47 | 23.15 | 24.11 |
| | L²QER | 30.83 | 51.19 | 60.01 | 36.18 | 40.95 | 26.62 | 43.41 | 52.57 | 26.4 | 42.17 | 25.02 | 26.68 | 25.41 | 25.88 | 25.67 |
| | L²QER-MXFP4 | 13.78 | 61.44 | 69.26 | 42.32 | 53.91 | 30.03 | 55.06 | 55.25 | 33 | 50.22 | 26.7 | 28.68 | 26.86 | 25.15 | 27.1 |
| | SERQ (RTN) | 13.57 | 57.68 | 69.26 | 40.28 | 55.93 | 32.94 | 56.63 | 55.25 | 33.2 | 50.15 | 28.78 | 30.51 | 29.64 | 27.69 | 29.11 |
| | SERQ (GPTQ) | 12.52 | 59.85 | 70.13 | 40.58 | 53.41 | 32.68 | 57.84 | 54.62 | 34.4 | 50.44 | 26.93 | 27.84 | 26.13 | 24.2 | 26.34 |
| | SERQ-MXFP4 | 13.71 | 60.03 | 68.77 | 40.58 | 54.08 | 31.83 | 55.34 | 56.67 | 32.2 | 49.94 | 27.55 | 28.23 | 27.49 | 25.5 | 27.23 |

**LLaMA-3.2-3B**

| Model | Methods | PPL (↓) | BoolQ (↑) | PIQA (↑) | SIQA (↑) | ARC-e (↑) | ARC-c (↑) | HellaS. (↑) | WinoG. (↑) | OBQA (↑) | Avg. (↑) | Human. (↑) | Other (↑) | SocialS. (↑) | STEM (↑) | Avg. (↑) |
|---|---|---|---|---|---|---|---|---|---|---|---|---|---|---|---|---|
| LLaMA-3.2-3B | baseline | 7.81 | 72.97 | 77.53 | 46.93 | 71.59 | 46.25 | 73.49 | 69.53 | 43 | 62.66 | 48.78 | 63.08 | 62.59 | 44.72 | 54.06 |
| | LLM.int4() | 4.37e+2 | 45.41 | 52.67 | 33.93 | 28.07 | 22.61 | 27.94 | 49.49 | 26 | 35.77 | 25.02 | 24.27 | 22.07 | 21.88 | 23.5 |
| | L²QER | 14.11 | 57.37 | 69.21 | 40.53 | 53.28 | 34.04 | 60.73 | 58.17 | 34 | 50.92 | 30.44 | 34.63 | 33.99 | 30.89 | 32.25 |
| | L²QER-MXFP4 | 9.59 | 67.92 | 74.86 | 44.78 | 65.57 | 41.47 | 68.37 | 63.93 | 40.4 | 58.41 | 42.47 | 54.3 | 52.68 | 40.37 | 46.85 |
| | SERQ (RTN) | 9.43 | 67.74 | 73.72 | 45.45 | 66.33 | 42.32 | 69.09 | 64.96 | 37.6 | 58.4 | 43.06 | 53.91 | 51.09 | 39.23 | 46.36 |
| | SERQ (GPTQ) | 9.15 | 67.52 | 74.48 | 45.29 | 64.73 | 42.06 | 69.57 | 66.38 | 38 | 58.5 | 42.27 | 53.97 | 50.37 | 38.09 | 45.7 |
| | SERQ-MXFP4 | 8.74 | 68.56 | 73.83 | 45.24 | 61.83 | 40.02 | 67.16 | 64.33 | 39 | 57.5 | 38.68 | 47.89 | 45.82 | 34.44 | 41.33 |

**Qwen-2.5-3B**

| Model | Methods | PPL (↓) | BoolQ (↑) | PIQA (↑) | SIQA (↑) | ARC-e (↑) | ARC-c (↑) | HellaS. (↑) | WinoG. (↑) | OBQA (↑) | Avg. (↑) | Human. (↑) | Other (↑) | SocialS. (↑) | STEM (↑) | Avg. (↑) |
|---|---|---|---|---|---|---|---|---|---|---|---|---|---|---|---|---|
| Qwen-2.5-3B | baseline | 8.03 | 77.58 | 78.73 | 50.1 | 73.11 | 47.1 | 73.62 | 68.51 | 41.8 | 63.82 | 56.77 | 71.07 | 76.15 | 60.96 | 65.12 |
| | LLM.int4() | 8.51e+5 | 51.96 | 50.49 | 34.19 | 24.58 | 25.09 | 26.52 | 49.96 | 30.6 | 36.67 | 24.12 | 23.66 | 21.9 | 21.63 | 22.97 |
| | L²QER | 221.5 | 40.09 | 56.91 | 35.06 | 35.44 | 25.17 | 33.78 | 51.93 | 28.6 | 38.37 | 24.65 | 24.91 | 22.56 | 22.68 | 23.81 |
| | L²QER-MXFP4 | 10.46 | 72.51 | 75.84 | 47.13 | 73.4 | 45.31 | 67.21 | 64.25 | 38.8 | 60.56 | 49.8 | 61.25 | 64.61 | 51.09 | 55.87 |
| | SERQ (RTN) | 9.66 | 72.6 | 75.63 | 49.03 | 72.14 | 44.11 | 70 | 64.64 | 41.8 | 61.24 | 52.52 | 65.88 | 69.97 | 53.5 | 59.52 |
| | SERQ (GPTQ) | 9.35 | 74.74 | 75.73 | 45.55 | 72.14 | 45.31 | 69.92 | 63.22 | 39.6 | 60.78 | 52.54 | 65.92 | 70.43 | 53.12 | 59.55 |
| | SERQ-MXFP4 | 9.79 | 75.17 | 74.81 | 46.88 | 69.57 | 44.28 | 68.85 | 63.85 | 40 | 60.43 | 49.8 | 61.92 | 64.9 | 51.32 | 56.13 |

Table 13: Accuracy results of distribution flattening methods when tested with W4A4 settings.

| Model | Methods | PPL (↓) | 0-Shot Common Sense Reasoning tasks | | | | | | | | | MMLU | | | | |
|---|---|---|---|---|---|---|---|---|---|---|---|---|---|---|---|---|
| | | | BoolQ (↑) | PIQA (↑) | SIQA (↑) | ARC-e (↑) | ARC-c (↑) | HellaS. (↑) | WinoG. (↑) | OBQA (↑) | Avg. (↑) | Human. (↑) | Other (↑) | SocialS. (↑) | STEM (↑) | Avg. (↑) |
| LLaMA-2-7B | baseline | 5.47 | 77.74 | 79.05 | 46.11 | 74.49 | 46.25 | 76 | 68.9 | 44.2 | 64.09 | 39.72 | 47.12 | 47.45 | 34.28 | 41.83 |
| | SmoothQ | 1.51e+4 | 44.16 | 49.95 | 34.49 | 25.97 | 27.82 | 26.39 | 47.51 | 27.2 | 35.44 | 25.48 | 22.98 | 29.48 | 26.55 | 26.04 |
| | SmoothQ(g128) | 7.49 | 68.59 | 74.48 | 41.45 | 65.4 | 39.85 | 67.99 | 61.64 | 37.8 | 57.15 | 28.63 | 31.9 | 32.92 | 29.12 | 30.4 |
| | QuaRot | 6.15 | 72.84 | 77.2 | 33.06 | 71.59 | 43 | 72.3 | 64.64 | 41.6 | 59.53 | 32.52 | 36.85 | 36.72 | 28.86 | 33.58 |
| | SpinQuant | 6 | 73.8 | 76 | 44.1 | 43.6 | 71.3 | 73.2 | 65.4 | 40.4 | 61 | 33.9 | 38.5 | 37.5 | 29.5 | 34.8 |
| | SERQ-MXFP4 | 6.22 | 73 | 77.2 | 44.17 | 70.2 | 43.69 | 72.59 | 68.19 | 41 | 61.26 | 32.16 | 40.01 | 39.1 | 31.43 | 35.25 |

| Model | Methods | PPL (↓) | 0-Shot Common Sense Reasoning tasks | | | | | | | | | MMLU | | | | |
|---|---|---|---|---|---|---|---|---|---|---|---|---|---|---|---|---|
| | | | BoolQ (↑) | PIQA (↑) | SIQA (↑) | ARC-e (↑) | ARC-c (↑) | HellaS. (↑) | WinoG. (↑) | OBQA (↑) | Avg. (↑) | Human. (↑) | Other (↑) | SocialS. (↑) | STEM (↑) | Avg. (↑) |
| LLaMA-2-13B | baseline | 4.88 | 80.61 | 80.52 | 47.39 | 77.53 | 49.23 | 79.38 | 72.38 | 45.2 | 66.53 | 47.89 | 59.29 | 61.16 | 42.21 | 52.04 |
| | SmoothQ | 1.32e+4 | 38.47 | 49.24 | 34.7 | 26.81 | 27.73 | 25.72 | 47.75 | 25.8 | 34.53 | 24.48 | 26.62 | 23.4 | 23.85 | 23.85 |
| | SmoothQ(g128) | 6.31 | 75.23 | 76.93 | 43.19 | 70.5 | 44.45 | 72.52 | 68.43 | 39 | 61.28 | 35.9 | 44.13 | 45.76 | 35.68 | 39.83 |
| | QuaRot | 5.41 | 78.47 | 78.89 | 33.32 | 73.7 | 46.25 | 76.29 | 70.48 | 43 | 62.55 | 43.68 | 52.85 | 55.41 | 39.11 | 47.25 |
| | SpinQuant | 5.2 | 78.2 | 79.3 | 46.3 | 49 | 76.3 | 77.1 | 69.5 | 42.8 | 64.8 | 43.5 | 53.1 | 55.4 | 39.1 | 47.8 |
| | SERQ-MXFP4 | 5.39 | 77.86 | 78.73 | 45.19 | 75.59 | 48.12 | 76.18 | 70.09 | 43 | 64.35 | 43.59 | 52.33 | 54.6 | 40.28 | 47.19 |

| Model | Methods | PPL (↓) | 0-Shot Common Sense Reasoning tasks | | | | | | | | | MMLU | | | | |
|---|---|---|---|---|---|---|---|---|---|---|---|---|---|---|---|---|
| | | | BoolQ (↑) | PIQA (↑) | SIQA (↑) | ARC-e (↑) | ARC-c (↑) | HellaS. (↑) | WinoG. (↑) | OBQA (↑) | Avg. (↑) | Human. (↑) | Other (↑) | SocialS. (↑) | STEM (↑) | Avg. (↑) |
| LLaMA-3-8B | baseline | 6.13 | 81.07 | 80.74 | 47.08 | 77.69 | 52.99 | 79.2 | 73.48 | 45 | 67.16 | 54.81 | 70.55 | 73.25 | 53.89 | 62.13 |
| | SmoothQ | 4.71e+5 | 50.61 | 51.69 | 33.78 | 24.2 | 26.02 | 26.65 | 49.57 | 28.2 | 36.34 | 26.65 | 24.36 | 24.21 | 24.71 | 25.17 |
| | SmoothQ(g128) | 17.26 | 61.83 | 64.2 | 39.92 | 46.59 | 30.38 | 55.61 | 59.59 | 33.6 | 48.97 | 29.33 | 30.16 | 29.61 | 28.1 | 29.3 |
| | QuaRot | 8.41 | 70.49 | 77.04 | 32.96 | 69.57 | 43.26 | 72.22 | 64.64 | 42.8 | 59.12 | 42.76 | 53.56 | 53.56 | 41.74 | 47.29 |
| | SpinQuant | 8.26 | 73.4 | 75.2 | 44.4 | 72 | 46.9 | 71.9 | 67.7 | 42.4 | 61.75 | 45.8 | 56.5 | 57.2 | 42.5 | 49.93 |
| | SERQ-MXFP4 | 7.63 | 76.15 | 77.2 | 44.11 | 72.69 | 45.39 | 75.13 | 68.43 | 42.6 | 62.71 | 47.72 | 60.7 | 63.18 | 46.84 | 53.48 |

| Model | Methods | PPL (↓) | 0-Shot Common Sense Reasoning tasks | | | | | | | | | MMLU | | | | |
|---|---|---|---|---|---|---|---|---|---|---|---|---|---|---|---|---|
| | | | BoolQ (↑) | PIQA (↑) | SIQA (↑) | ARC-e (↑) | ARC-c (↑) | HellaS. (↑) | WinoG. (↑) | OBQA (↑) | Avg. (↑) | Human. (↑) | Other (↑) | SocialS. (↑) | STEM (↑) | Avg. (↑) |
| LLaMA-3.2-1B | baseline | 9.75 | 63.67 | 74.48 | 42.84 | 60.44 | 36.18 | 63.73 | 59.83 | 37.4 | 54.82 | 34.92 | 41.1 | 39.78 | 32.29 | 36.76 |
| | SmoothQ | 1.53e+5 | 39.14 | 51.8 | 32.7 | 26.35 | 28.16 | 25.44 | 50.12 | 31 | 35.59 | 24.59 | 24.46 | 24.89 | 23.44 | 24.37 |
| | SmoothQ(g128) | 69.22 | 51.62 | 56.09 | 35.26 | 36.99 | 26.11 | 37.9 | 49.33 | 27 | 40.04 | 24.65 | 26.1 | 23.24 | 23.63 | 24.43 |
| | QuaRot | 13.17 | 60.21 | 69.7 | 39.92 | 54.17 | 32.17 | 55.65 | 55.25 | 33.2 | 50.03 | 26.55 | 27.62 | 25.71 | 26.74 | 26.64 |
| | SpinQuant | 13.47 | 60.1 | 68.8 | 39 | 50.2 | 30.6 | 55.4 | 55.5 | 32.2 | 48.95 | 26.4 | 27.7 | 26 | 25.5 | 26.38 |
| | SERQ-MXFP4 | 13.71 | 60.03 | 68.77 | 40.58 | 54.08 | 31.83 | 55.34 | 56.67 | 32.2 | 49.94 | 27.55 | 28.23 | 27.49 | 25.5 | 27.23 |

| Model | Methods | PPL (↓) | 0-Shot Common Sense Reasoning tasks | | | | | | | | | MMLU | | | | |
|---|---|---|---|---|---|---|---|---|---|---|---|---|---|---|---|---|
| | | | BoolQ (↑) | PIQA (↑) | SIQA (↑) | ARC-e (↑) | ARC-c (↑) | HellaS. (↑) | WinoG. (↑) | OBQA (↑) | Avg. (↑) | Human. (↑) | Other (↑) | SocialS. (↑) | STEM (↑) | Avg. (↑) |
| LLaMA-3.2-3B | baseline | 7.81 | 72.97 | 77.53 | 46.93 | 71.59 | 46.25 | 73.49 | 69.53 | 43 | 62.66 | 48.78 | 63.08 | 62.59 | 44.72 | 54.06 |
| | SmoothQ | 3.73e+4 | 40.46 | 51.52 | 33.32 | 25.34 | 26.11 | 26.38 | 51.46 | 31.2 | 35.72 | 23.91 | 23.53 | 23.33 | 22.61 | 23.41 |
| | SmoothQ(g128) | 53.33 | 53.7 | 62.51 | 37.41 | 42.89 | 27.99 | 49.04 | 53.2 | 26.6 | 44.17 | 27.38 | 27.81 | 27.62 | 26.42 | 27.31 |
| | QuaRot | 9.73 | 66.15 | 73.34 | 43.45 | 59.81 | 37.29 | 68.12 | 60.85 | 37.2 | 55.76 | 41.23 | 51.05 | 50.76 | 37.93 | 44.75 |
| | SpinQuant | 10.15 | 68.5 | 73 | 43.2 | 62.8 | 38.9 | 67.6 | 63.1 | 38 | 56.88 | 39.9 | 47.1 | 47.7 | 36.4 | 42.42 |
| | SERQ-MXFP4 | 8.74 | 68.56 | 73.83 | 45.24 | 61.83 | 40.02 | 67.16 | 64.33 | 39 | 57.5 | 38.68 | 47.89 | 45.82 | 34.44 | 41.33 |

