# OpenReview forum: "SERQ: Saliency-Aware Low-Rank Error Reconstruction for LLM Quantization"
_ICLR.cc/2026/Conference — ICLR 2026 Poster_

### Official Review · Reviewer_3ZBf · 2025-10-28

**Soundness:** 3
**Presentation:** 3
**Contribution:** 2
**Rating:** 4
**Confidence:** 4

**Summary:**

This paper introduces SERQ. While existing low-rank adaptation methods are effective, they often face severe accuracy degradation in low-bit W4A4 (4-bit weight, 4-bit activation) settings and require two sequential factors, which limits inference efficiency. SERQ addresses this by employing a single low-rank compensation matrix to jointly mitigate quantization errors arising from both activation and weight saliency. The method operates in three stages: (1) static activation flattening, (2) saliency-aware error reconstruction, and (3) offline weight permutation. This design preserves efficient 4-bit matrix multiplication, keeps latency overhead minimal by performing most operations offline , and empirically outperforms prior error reconstruction and state-of-the-art rotation-based W4A4 approaches.

**Strengths:**

1. The motivation of this paper is clear, quantizing both main branch and lora branch to W4A4, as well as reducing the overhead of lora branch through saliency channel detection.
2. The ablation studies is comprehensive.

**Weaknesses:**

1. More recent state-of-the-art baselines—such as DuQuant [1], OSTQuant[2] and FlatQuant[3] for W4A4KV4 quantization—should be included for comparison.
2. SVDQuant [4] also use channel-wise scaling (static activation flattening in this paper) and LoRA error compensate, though it was used in diffusion model. This paper should disscuss more about such similar work.
3. End-to-end prefilling and decoding speedup should be included to demonstrate the efficiency of proposed method.
4. The comparison results in Table 2 seems unfair. Flattening methods and proposed SEQR use different quantization formats. Additionally, the SEQR-MXFP4 average bit-width is higher than QuaRot and SpinQuant.
[1]  DuQuant: Distributing Outliers via Dual Transformation Makes Stronger Quantized LLMs, NeurIPS 2024
[2] OSTQuant: Refining Large Language Model Quantization with Orthogonal and Scaling Transformations for Better Distribution Fitting, ICLR 2025
[3] FlatQuant: Flatness Matters for LLM Quantization, ICML 2025
[4] SVDQUANT: ABSORBING OUTLIERS BY LOW-RANK COMPONENTS FOR 4-BIT DIFFUSION MODELS, ICLR 2025

**Questions:**

What quantization format is used for flattening methods in Table 2?

---

> ### Author Response · Authors · 2025-11-21
> **Part 1**
>
> We are truly grateful for your detailed review. We have deeply considered your feedback and have done our best to address each of your concerns in the response below.
>
> > **W1: More recent state-of-the-art baselines—such as DuQuant [1], OSTQuant[2] and FlatQuant[3] for W4A4KV4 quantization—should be included for comparison.**
> ---
> We appreciate this constructive comment. We fully agree with your suggestion that comparing our method against recent rotation-based state-of-the-art (SOTA) approaches is necessary to strengthen our evaluation. Our work primarily focuses on LoRA-based quantization, proposing a novel scheme that achieves W4A4 quantization for both weights and activations while minimizing the latency overhead in the low-rank path. While we initially prioritized comparisons within the LoRA-based quantization, we acknowledge that our comparison with the latest rotation-based methods, which represent the SOTA in W4A4 quantization, was insufficient.
> Accordingly, we have conducted additional experiments to include the baselines you suggested (DuQuant, OSTQuant, and FlatQuant) (see **Appendix A.5** and **Table 9**). For brevity, we present the results for the LLaMA-2 7B model (selected from our full evaluation on LLaMA-2 7B/13B and LLaMA-3.2 3B). The results are summarized as follows:
>
> | Model | Method | Calib Time | PPL | 0-shot | MMLU |
> | --- | :--- | --- | --- | --- | --- |
> | LLaMA-2 7B | FP16 | - | 5.47 | 64.09 | 41.83 |
> | | SpinQuant | $\sim$598m | 6 | 61 | 34.8 |
> | | FlatQuant | $\sim$131m | **5.75** | **62.11** | **38.24** |
> | | DuQuant | $\sim$255m | 5.95 | 61.84 | 33.46 |
> | | OSTQuant | $\sim$72m | 5.91 | 61.72 | 37.39 |
> | | QuaRot | $\sim$31m | 6.15 | 59.53 | 33.58 |
> | | SERQ | **$\sim$23m** | 5.97 | 61.87 | 37.03 |
>
> The results indicate that methods employing extensive training or heavy iterative optimization generally yield the highest accuracy. FlatQuant, in particular, demonstrates superior performance, outperforming all the compared models. However, it is important to note that this advantage comes at the expense of optimizing a large number of trainable parameters and introducing floating-point (FP) affine transformations during runtime. Except for FlatQuant, all other methods including SERQ do not employ high-precision GEMM in their linear layer operations. Regarding calibration efficiency, training-free approaches (specifically SERQ and QuaRot) offer drastically lower calibration costs. What is particularly compelling is that, despite this negligible overhead, SERQ delivers performance that fully comparable to computationally intensive, training-based methods. This confirms that SERQ acts as a highly efficient, lightweight framework, offering an optimal trade-off between calibration cost and model accuracy.
>
> &nbsp;
>
> > **W2: SVDQuant also use channel-wise scaling (static activation flattening in this paper) and LoRA error compensate, though it was used in diffusion model. This paper should disscuss more about such similar work.**
> ---
> We sincerely appreciate this valuable pointer. First of all, we apologize for overlooking SVDQuant, which shares similarities with our work in utilizing per-channel scaling and LoRA-based error compensation. We fully acknowledge that our discussion of related works was insufficient. To address this, we have established a new related works section in **Appendix A.2**, where we comprehensively discuss various relevant studies.
>
> Here, we highlight the key distinctions between our method and the works involving channel-wise scaling and LoRA error compensation:
>
> * SVDQuant: While SVDQuant is similar in that it employs channel-wise scaling and LoRA for error compensation, there are two primary differences. First, as you correctly noted, it targets diffusion models, whereas our work focuses on Large Language Models (LLMs), addressing different domain-specific challenges. Second, regarding the low-rank path, SVDQuant maintains two separate layers in full-precision(SERQ is pure-4bits).
> - QERA[1]: This work contributes a closed-form analytical method to precisely reconstruct weight quantization errors, eliminating heuristics found in prior works, such as LQER[2]. However, it diverges from our work in two ways. It focuses only on weight quantization, and utilizes two layers with 8-bit quantization for the low-rank path.
> - ASER[3]: ASER performs low-rank decomposition reflecting activation distributions via whitening SVD and also employs both channel-wise scaling and LoRA compensation. The key difference lies in structure. ASER still relies on two low-rank matrices and employs mixed precision (e.g., W4A8 or W4A6).
>
> For a more detailed discussion on other related works, we respectfully invite you to refer to **Appendix A.2.**
>
> [1] QERA: an Analytical Framework for Quantization Error Reconstruction.
>
> [2] LQER: Low-Rank Quantization Error Reconstruction for LLMs.
>
> [3] ASER: Activation Smoothing and Error Reconstruction for Large Language Model Quantization.

---

> ### Author Response · Authors · 2025-11-21
> **Part 2**
>
> > **W3: End-to-end prefilling and decoding speedup should be included to demonstrate the efficiency of proposed method.**
> ---
> We appreciate this valuable suggestion. While our original manuscript focused on analyzing the latency overhead for the Llama-3-8B model at the decoder layer level, we fully agree that end-to-end performance metrics are essential to demonstrate efficiency in real-time scenarios.
>
> Accordingly, we measured the end-to-end performance using custom MXFP quantization kernels and CUTLASS MXFP4 GEMM kernels. Note that structurally, SERQ only adds a single narrow MXFP GEMM to the linear operation.
>
> | Bsz | Method | TTFT(ms) | Speedup | Peak Mem(GB) | Saving | TPOT(ms) | Speedup |
> | --- | :--- | --- | --- | --- | --- | --- | --- |
> | 1 | FP16 | 132.38 | $\times$1 | 17.44 |  $\times$1 | 18.44 | $\times$1 |
> | | MXFP4 | 56.03 | $\times$2.36 | 6.94 | $\times$2.51 | 25.48 | $\times$0.72 |
> | | SERQ-MXFP4 | 62.31 | $\times$2.12 | 7.03 | $\times$2.48 | 39.28 | $\times$0.47 |
> | 8 | FP16 | 1252.24 | $\times$1 | 27.9 |  $\times$1 | 61.93 | $\times$1 |
> | | MXFP4 | 542.58 | $\times$2.31 | 17.4 | $\times$1.6 | 54.49 | $\times$1.14 |
> | | SERQ-MXFP4 | 587.17 | $\times$2.13 | 17.48 | $\times$1.6 | 57.08 | $\times$1.08 |
> | 16 | FP16 | 2515.08 | $\times$1 | 39.85 |  $\times$1 | 114.24 | $\times$1 |
> | | MXFP4 | 1113.06 | $\times$2.26 | 29.35 | $\times$1.36 | 100.71 | $\times$1.13 |
> | | SERQ-MXFP4 | 1206.69 | $\times$2.08 | 29.44 | $\times$1.35 | 103.27 | $\times$1.11 |
> | 32 | FP16 | 4987.05 | $\times$1 | 63.76 |  $\times$1 | 218.01 | $\times$1 |
> | | MXFP4 | 2257.54 | $\times$2.21 | 53.25 | $\times$1.2 | 193.7 | $\times$1.13 |
> | | SERQ-MXFP4 | 2453.29 | $\times$2.03 | 53.34 | $\times$1.2 | 202.13 | $\times$1.08 |
>
> * Prefilling: For the compute-intensive prefilling phase, our method consistently achieves a ~2x speedup compared to the FP16 baseline across various batch sizes. Notably, SERQ introduces a negligible latency overhead of approximately less than 10% compared to raw MXFP4 GEMM, while maintaining a very low memory footprint due to the small number of 4-bit low-rank parameters.
> * Decoding: We observe that at low batch sizes, MXFP4 can be slower than FP16. This is because the fine-grained structure (small quantization groups) of the MX-format incurs overhead that dominates during the decoding of small batches. However, this effect diminishes as the batch size increases. At larger batch sizes (8-32), MXFP4 and our method surpass FP16 performance.
>
> Ultimately, SERQ demonstrates its practical utility in real-time inference by delivering significant gains in prefilling speed and memory savings, with only minimal latency overhead compared to  MXFP.
>
> &nbsp;
>
> > **W4: The comparison results in Table 2 seems unfair. Flattening methods and proposed SEQR use different quantization formats. Additionally, the SEQR-MXFP4 average bit-width is higher than QuaRot and SpinQuant.**
> ---
> We appreciate this critical observation regarding fair comparison. Regarding the format discrepancy,  we Initially presented MX-format results in **Table 2** to align with the latest hardware trends, as architectures like NVIDIA Blackwell are moving away from native 4-bit integers. However, we fully acknowledge that this created a format mismatch. While our main concepts are fundamentally rooted in integer quantization (SERQ-RTN and GPTQ are all INT-based), we have revised **Table 2** to use a unified Integer format for a strictly fair comparison. By switching from MX to Integer format, the effective bit-width decreased from 4.37 to 4.24.
>
> Surprisingly, despite the reduced bit budget, our method exhibits superior performance, surpassing the baselines. This demonstrates that our performance gains stem from the algorithmic effectiveness of our approach rather than simply using more bits. We acknowledge that due to the additional LoRA parameters, our effective bit-width (4.24) remains slightly higher than pure 4-bit methods.
>
> Although our primary focus lies in LoRA-based quantization, given that our method achieves full W4A4 capability, we deemed it necessary to benchmark against Rotation-based methods, which represent the current SOTA in this regime. Consistent with the slightly higher effective bit-width, our method demonstrates superior performance across the majority of tasks. Furthermore, it ensures deterministic (static) performance without internal randomnes (e.g., QuaRot), and  features lightweight calibration(see **Table 9**) and high adaptability. Therefore, we argue that SERQ presents a more efficient and practical quantization framework.
>
> &nbsp;
>
> ---
> We sincerely thank you again for the time and effort you dedicated to reviewing our paper. We have done our very best to address your concerns through detailed explanations and additional experiments. We hope that our response has satisfactorily addressed your questions and that you will consider our revision favorably.
>
> Sincerely, The Authors

---

### Official Review · Reviewer_oK6e · 2025-10-29

**Soundness:** 2
**Presentation:** 2
**Contribution:** 2
**Rating:** 4
**Confidence:** 4

**Summary:**

The paper proposes SERQ, a method for 4-bit quantization of large language models that uses a single low-rank compensation matrix to correct quantization errors. The approach combines three components: (1) static activation flattening via SmoothQuant-style scaling, (2) saliency-aware error reconstruction using only the most significant weight rows, and (3) offline weight permutation to avoid runtime overhead. The authors claim their method enables true end-to-end 4-bit computation while outperforming prior low-rank error reconstruction methods like L2QER and rotation-based approaches like QuaRot and SpinQuant.

**Strengths:**

● Novel Design for Latency Reduction: The paper proposes a scheme that combines a single compensation matrix with an offline permutation, with the goal of eliminating the latency overhead found in conventional two-factor error correction methods.
● Presents Experimental Results on Multiple Models: The paper reports experimental results across several modern LLMs and includes latency measurements on recent hardware to support its efficiency claims.
● Addresses a Significant Problem: The work targets the challenging and important problem of W4A4 quantization, which is of high interest to the community for efficient LLM deployment.

**Weaknesses:**

While the paper presents some interesting ideas, it suffers from significant flaws in its theoretical grounding, experimental validation, and practical considerations, which severely undermine the credibility and value of its contributions.
1. Critically Flawed and Incomplete Experimental Validation
● Flawed Experimental Comparisons: The paper's main results are built on unfair comparisons. For instance, SERQ is allocated a higher bit-budget (4.37 bits) than its main competitors (~4 bits), which may in itself account for its accuracy advantage. Furthermore, the performance of the key L2QER baseline is suspiciously low on certain models, casting doubt on the entire set of comparative results.
● Critically Limited Scope and Unproven Generalization: The paper’s experimental validation is fundamentally disconnected from modern, real-world LLM scenarios. The evaluation is confined to small-scale (<13B), dense models, failing to address three critical dimensions: (1) Scale: It lacks validation on large-scale models (70B+), where quantization challenges are most severe. (2) Architecture: It provides no evidence on diverse, prevalent architectures like Mixture-of-Experts (MoE), where its core saliency heuristic may not hold. (3) Context Length: It fails to evaluate performance in long-context scenarios (e.g., 8K-32K), which are vital for modern applications and where quantization errors are known to accumulate. This critically narrow scope makes it impossible to assess the method's true scalability and practical utility.
2. Dismissal of Practical Issues and Lack of Transparency
● Dismissal of Practical Drawbacks: The paper touts its offline permutation as a "zero-latency" solution while completely ignoring the immense software engineering challenges and loss of modularity it introduces, especially concerning its interaction with distributed inference techniques like tensor parallelism. This dismissal of practical drawbacks makes the method's utility questionable.
● Missing Analysis of Quantization Cost: The paper completely omits an analysis of its quantization cost. As a Post-Training Quantization (PTQ) method, the time and compute resources required for calibration are critical efficiency metrics. The absence of this data prevents a full assessment of its overall cost-effectiveness.
● Poor Explanation of Key Components: The paper fails to clearly explain how its best-performing variant (SERQ-GPTQ) works. The description in the appendix is contradictory and confusing, which is a major reporting weakness for a paper that claims top results with this variant and impacts its reproducibility.
3. Weak Theoretical Underpinnings and Incremental Contribution
● Incremental Contribution and Overstated Novelty: The method is essentially a patchwork of existing techniques (SmoothQuant, AWQ's saliency metric, LoRA-style correction). Its primary novel component—the offline permutation—is more of an engineering trick than a fundamental algorithmic breakthrough. The paper conveniently ignores the incremental nature of its contribution and overstates its novelty.
● Limited Theoretical Justification for Saliency Metric: The method relies entirely on a heuristic for saliency borrowed from AWQ (identifying rows by activation scale). The paper fails to provide any theoretical justification as to why this simple statistical metric is optimal for error reconstruction, nor does it rigorously compare it against other potential channel selection paradigms.

**Questions:**

Please address my concerns in Weaknesses.

---

> ### Author Response · Authors · 2025-11-21
> **Part 1**
>
> First of all, we would like to sincerely thank you for the time and effort you dedicated to reviewing our paper. We have done our very best to address the concerns you raised in the following response.
> > **W1-1: Flawed Experimental Comparisons: The paper's main results are built on unfair comparisons. For instance, SERQ is allocated a higher bit-budget (4.37 bits) than its main competitors (~4 bits), which may in itself account for its accuracy advantage. Furthermore, the performance of the key L2QER baseline is suspiciously low on certain models, casting doubt on the entire set of comparative results.**
> ---
> We appreciate this critical observation. While our initial use of the MX-format aimed to demonstrate compatibility with emerging hardware like NVIDIA Blackwell, we acknowledge that it complicated the comparison. Given that our core concepts are rooted in integer quantization(both SERQ-RTN and SERQ-GPTQ are integer-based), we have revised **Table 2** to standardize all results using the Integer format.
>
> With this transition, our effective bit-width decreased from 4.37 to 4.24 bits(an inherent characteristic of LoRA-based approaches). While this remains marginally higher than pure 4-bit methods, we deemed it necessary to benchmark against SOTA Rotation-based methods given our full W4A4 capability. Consistent with the slightly higher bit-width, our method demonstrates superior performance across the majority of tasks. Moreover, unlike Rotation-based approaches (e.g., QuaRot), SERQ ensures deterministic performance free from internal randomness, while featuring lightweight calibration and high adaptability. Consequently, we believe SERQ offers a highly efficient and practical quantization framework.
>
> Regarding the L$^2$QER baseline, we understand your concern about its low W4A4 performance. It is important to note that L$^2$QER was originally designed for mixed precision (W4A8 or W4A6). Strictly enforcing 4-bit activation quantization without algorithmic modification caused the observed performance drop, whereas it performs reasonably under its original settings. We rigorously double-checked these experiments during the rebuttal and confirmed consistent results under the W4A4 constraint. We hope this clarifies the validity of our baselines.
>
> &nbsp;
>
> > **W1-2: Critically Limited Scope and Unproven Generalization: The paper’s experimental validation is fundamentally disconnected from modern, real-world LLM scenarios. The evaluation is confined to small-scale (<13B), dense models, failing to address three critical dimensions: (1) Scale: It lacks validation on large-scale models (70B+), where quantization challenges are most severe. (2) Architecture: It provides no evidence on diverse, prevalent architectures like Mixture-of-Experts (MoE), where its core saliency heuristic may not hold. (3) Context Length: It fails to evaluate performance in long-context scenarios (e.g., 8K-32K), which are vital for modern applications and where quantization errors are known to accumulate. This critically narrow scope makes it impossible to assess the method's true scalability and practical utility.**
> ---
> We understand your concern regarding the experimental scope. Our initial evaluation primarily followed the experiments common in prior quantization studies. To fully address your concerns, we have organized our response by category, starting with Scalability.
>
> (1) Scalability: While our original manuscript focused on models under 13B, we acknowledge the necessity of testing on larger-scale models (70B+). To demonstrate the suitability of our method for modern high-performance LLMs, we conducted additional experiments using **Llama-2-70B**. These new results have been incorporated into **Table 1** of the revised manuscript. and the results are as follows:
>
> | Model | Bits | Method | PPL | 0-shot | MMLU |
> | --- |  --- |  :--- |  --- |  --- |  --- |
> | LLaMA-2 70B | FP16 | baseline | 3.53 | 70.54 | 65.44 |
> | | W4A8 | LLM.int4() | 7.41e+3 | 49.71 | 43.56 |
> | | | L$^2$QER | 3.55 | 69.78 | 64.45 |
> | | | SERQ (RTN) | 3.44 | **70.35** | 64.45 |
> | | | SERQ (GPTQ) | **3.43** | 70.31 | **64.62**|
> | | W4A4 | LLM.int4() | 2.76e+4 | 39.77 | 23.67 |
> | | | L$^2$QER | 4.55 | 66.92 | 56.44 |
> | | | L$^2$QER-MXFP4 | 3.81 | 69.03 | 61.48 |
> | | | SERQ (RTN) | 3.65 | **69.68** | 63.15 |
> | | | SERQ (GPTQ) | **3.64** | 69.38 | 63.09 |
> | | | SERQ-MXFP4 | 3.79 | 69.58 | 61.64 |
>
> As the results demonstrate, our method consistently achieves state-of-the-art (SOTA) performance among LoRA-based quantization approaches, even at the 70B scale. We sincerely thank you for this suggestion, as including these large-scale experiments has significantly strengthened the generalization and empirical robustness of our work.

---

> > ### Author Response · Authors · 2025-11-21
> > **Part 2**
> >
> > (2) Architecture: We acknowledge that the original manuscript lacked a detailed discussion on the adaptability of our method across diverse architectures. We would like to clarify our advantages and address the MoE concern as follows.
> >
> > In Llama-like models, A key advantage of SERQ lies in its handling of the FFN gate projection. Conventional activation flattening methods often employ orthogonal transformations (rotations) to stabilize distributions. However, the SiLU activation function following the gate projection is non-linear and operates element-wise, which prevents the merging of these rotation matrices into the weights. This typically necessitates additional online computational modules, incurring overhead. In contrast, as illustrated in **Figure 2(b) ($P_4$)**, SERQ utilizes permutation. Since the permutation operation can be seamlessly propagated through element-wise functions like SiLU, they can be effectively merged into the preceding weights, eliminating runtime overhead.
> >
> > $$
> > SiLU(\underline{x}) \ P= \left( \underline{x} \cdot \left( {\begin{bmatrix} 1 & 1 & \cdots \\ \end{bmatrix}} + e^{-\underline{x}} \right)^{-1} \right) \ P \\ = (\underline{x} \ P ) \cdot \left({\begin{bmatrix} 1 & 1 & \cdots \\ \end{bmatrix}} + e^{-\underline{x} \cdot P}\right)^{-1} \\ = SiLU(\underline{x} \ P)
> > $$
> >
> > Furthermore, in models using Post-Norm (e.g., Gemma[1]), rotation-based methods face difficulties in merging rotations applied to the attention output. Building on the previously mentioned compatibility with element-wise operations, SERQ effectively bypasses this structural limitation.
> >
> > Regarding MoE model, we fully recognize the growing importance of MoE architectures (often used in multimodal contexts). Theoretically, since SERQ identifies salient weights based on activation distributions, it is intrinsically compatible with MoE structures, which rely on the same fundamental linear transformations as standard Transformers. We are currently actively configuring the experimental environment for MoE models. While establishing a robust setup requires some time, we are prioritizing this task and commit to updating the manuscript with the results immediately upon completion.
> >
> > &nbsp;
> >
> > (3) Context Length: We fully agree with your emphasis on the critical importance of long-context scenarios in modern LLM applications. Recognizing this, we actually included results for LongBench[2] (a benchmark for long-context generation) alongside GSM8K[3] in **Table 6**. However, we acknowledge that placing these results within the Ablation Study section likely reduced their visibility, which may have led to them being overlooked. To clarify, the LongBench datasets typically range from 4K to 32K. Specifically, we evaluated our method on QMSum, SAMSum, and RepoBench-P. The average context lengths for these datasets are detailed below.
> >
> > | Dataset | Avg len |
> > | :--- | :---: |
> > |QMSum | 10,614 |
> > |SAMSum| 6,258 |
> > |RepoBench-P | 4,206 |
> >
> > As demonstrated in **Table 6**, our method outperforms existing LoRA-based quantization techniques while showing minimal degradation compared to the FP16 baseline. This confirms that our approach maintains robust performance even in long-context scenarios. To improve visibility, we will highlight these results more prominently in the revised manuscript.
> >
> > [1] Gemma 2: Improving Open Language Models at a Practical Size.
> >
> > [2] LongBench: A Bilingual, Multitask Benchmark for Long Context Understanding.
> >
> > [3] Training Verifiers to Solve Math Word Problems.

---

> ### Author Response · Authors · 2025-11-21
> **Part 3**
>
> > **W2-2: Missing Analysis of Quantization Cost: The paper completely omits an analysis of its quantization cost. As a Post-Training Quantization (PTQ) method, the time and compute resources required for calibration are critical efficiency metrics. The absence of this data prevents a full assessment of its overall cost-effectiveness.**
> ---
> We sincerely appreciate this valuable suggestion. We fully agree that measuring the calibration cost is essential for a comprehensive assessment of Post-Training Quantization (PTQ) methods, where rapid deployment without extensive retraining is a key advantage. To address this, we measured the calibration time as a primary metric for quantization cost, comparing our method against relevant baselines(see **Table 9**). For brevity, we present the results for the LLaMA-2 7B model (selected from our full evaluation on LLaMA-2 7B/13B and -3.2 3B).
>
> | Model | Method | Calib Time | PPL | 0-shot | MMLU |
> | --- | :--- | --- | --- | --- | --- |
> | LLaMA-2 7B | FP16 | - | 5.47 | 64.09 | 41.83 |
> | | SpinQuant | $\sim$598m | 6 | 61 | 34.8 |
> | | FlatQuant | $\sim$131m | **5.75** | **62.11** | **38.24** |
> | | DuQuant | $\sim$255m | 5.95 | 61.84 | 33.46 |
> | | OSTQuant | $\sim$72m | 5.91 | 61.72 | 37.39 |
> | | QuaRot | $\sim$31m | 6.15 | 59.53 | 33.58 |
> | | SERQ | **$\sim$23m** | 5.97 | 61.87 | 37.03 |
>
> In this comparison, SERQ utilizes its main configuration with 128 calibration samples. For baselines such as SpinQuant, OSTQuant, FlatQuant, and DuQuant, we used the default settings provided in their official repositories, which typically involve iterative training or extensive optimization processes. As shown in the results, while methods involving extensive training or optimization generally achieve slightly higher accuracy, they incur significantly higher computational costs. FlatQuant, in particular, demonstrates superior performance. However, it is important to note that this advantage comes at the expense of optimizing a large number of trainable parameters and introducing floating-point (FP) affine transformations during runtime, which can incur additional computational overhead. Regarding calibration efficiency, training-free approaches (specifically SERQ and QuaRot) offer drastically lower calibration costs. What is particularly compelling is that, despite this negligible overhead, SERQ delivers performance that fully comparable to computationally intensive, training-based methods. This confirms the practical efficiency of our approach. We have included this detailed analysis in **Appendix A.5** of the revised manuscript.
>
> &nbsp;
>
> >**W2-3: Poor Explanation of Key Components: The paper fails to clearly explain how its best-performing variant (SERQ-GPTQ) works. The description in the appendix is contradictory and confusing, which is a major reporting weakness for a paper that claims top results with this variant and impacts its reproducibility.**
> ---
> We sincerely apologize for the confusion caused by the unclear description in the original manuscript. We fully agree that the explanation of SERQ-GPTQ was insufficient. We have completely rewritten the **Appendix A.3** in the revision to ensure much greater clarity. To briefly summarize the mechanism, Standard low-rank reconstruction often conflicts with GPTQ because GPTQ's iterative updates shift the weight distribution, causing it to diverge from the original statistics. To address this issue, we modified the pipeline to perform low-rank extraction prior to the GPTQ process.
>
> 1. We first extract high-variance components from salient rows into a 4-bit low-rank matrix (${R} = {Q}(\tilde{W}_s)$).
> 2. We then replace the original salient rows with the residual error ($\tilde{W}_s - R$).
> 3. Finally, GPTQ is applied to these stabilized residuals.
>
> This pre-extraction strategy effectively stabilizes the weight distribution and eliminates the dequantization-requantization process typically required for post-extraction, thereby making the main weights more amenable to GPTQ quantization.

---

> ### Author Response · Authors · 2025-11-21
> **Part 4**
>
> > **W3-1: Incremental Contribution and Overstated Novelty: The method is essentially a patchwork of existing techniques (SmoothQuant, AWQ's saliency metric, LoRA-style correction). Its primary novel component—the offline permutation—is more of an engineering trick than a fundamental algorithmic breakthrough. The paper conveniently ignores the incremental nature of its contribution and overstates its novelty.**
> ---
> We respectfully acknowledge your observation that our method builds upon foundational techniques such as SmoothQuant and AWQ. We view these works as foundational to the LLM PTQ field and have leveraged their insights. While we utilize established techniques, we do not simply concatenate them. Rather, we synergistically integrate them into a novel single-path low-rank architecture. This design allows us to:
>
> 1. Achieve a breakthrough in W4A4 quantization schemes within the LoRA-based quantization framework.
> 2. Optimize inference latency by maintaining only a single low-rank matrix, unlike prior works.
>
> Regarding the offline mergeable permutation, while it might be perceived as a mere engineering optimization, it represents a strategic algorithmic decision designed to significantly minimize  runtime overhead. We believe that this synergistic integration is precisely what enabled us to achieve stable, SOTA performance in the challenging W4A4 regime.
>
> &nbsp;
>
> > **W3-2: Limited Theoretical Justification for Saliency Metric: The method relies entirely on a heuristic for saliency borrowed from AWQ (identifying rows by activation scale). The paper fails to provide any theoretical justification as to why this simple statistical metric is optimal for error reconstruction, nor does it rigorously compare it against other potential channel selection paradigms.**
> ---
> We appreciate your constructive comments. We fully acknowledge that the theoretical justification in the original manuscript was insufficient. To address this, we have provided a comprehensive theoretical analysis supported by empirical evidence in the newly added **Appendix A.4** and **Table 8**. the results for **Table 8** are summarized below.
>
> | Model | Rank | Method | QSNR |
> | :---: | :---: | :--- | :---: |
> | LLaMA-3 8B | 128 | SVD | 19.87|
> | | | SERQ w/o SAF | 25.91 |
> | | 64 | SVD | 19.41 |
> | | | SERQ w/o SAF | 25.49 |
> | | 32 | SVD | 19.21 |
> | | | SERQ w/o SAF | 20.22 |
> | | 16 | SVD | 19.06 |
> | | | SERQ w/o SAF | 20.42 |
> | | 8 | SVD | 19.01 |
> | | | SERQ w/o SAF | 21.29 |
>
> Beyond these additions, we emphasize that the key factor enabling our method to achieve SOTA performance in LoRA-based quantization is the efficient utilization of the limited rank budget. Existing SVD-based methods typically aim to minimize the Frobenius-Norm of the quantization error by distributing the rank budget across the entire weight matrix. Crucially, when the low-rank path itself is quantized, these methods suffer from a compound loss—the error from rank reduction plus the error from quantization.
>
> In contrast, our method(SERQ) strategically allocates the limited rank budget to decompose salient weights without incurring rank reduction loss. Consequently, our approach suffers from only a single source of loss(the quantization error itself). As detailed in **Appendix A.4**, this combination of a specific saliency selection criterion and a quantization-efficient LoRA path demonstrates both the theoretical validity and practical effectiveness of our proposed method.
>
> We sincerely thank you for taking the time to read our detailed response. We hope that our clarifications have satisfactorily addressed your concerns.
>
> Sincerely, The Authors

---

> ### Comment · Reviewer_oK6e · 2025-11-25
>
> Thank you for providing such a detailed and comprehensive explanation in response to the questions raised. I will revisit and reconsider my previous score of this work.

---

> > ### Author Response · Authors · 2025-11-25
> >
> > Thank you very much for your prompt response. We hope our rebuttal has addressed your concerns, and we deeply appreciate your willingness to reconsider your evaluation of our work. Please feel free to ask for additional clarification. We remain fully available to engage in further discussion.

---

### Official Review · Reviewer_cLFy · 2025-10-31

**Soundness:** 3
**Presentation:** 4
**Contribution:** 3
**Rating:** 8
**Confidence:** 4

**Summary:**

This work proposes SERQ, a post-training quantization method that uses a single low-rank term to only compensate the error for salient weight rows in a pretrained linear layer, which further improves the inference throughput over baselines.

**Strengths:**

- The motivation is clear and the method is straightforward but effective
- The evaluation includes both model performance and hardware results

**Weaknesses:**

- The evaluation lacks model larger than 13B: all results were collected on model between 1B and 13B. It will be convincing if the author could offer results on larger models like 70B to verify the scalability of SERQ
- Details of runtime/hardware performance needs further clarification. Could the author elaborate more how a SERQ layer is accelerated on GPU? are there kernel fusions?

**Questions:**

Apart from the questions in Weakness section,

1. What does the "Quarot" block in Figure 2(b) mean? Is this for rotating activations?
2. SERQ is a heuristic based method. An ablation study of the static activation flattening seems needed here.
    - SmoothQuant was originally proposed for INT8 quantization, while SERQ uses MXFP.
    - In line 262, "However, unlike standalone prior approaches, the combined use of low-rank reconstruction enables effective compensation for the induced weight errors.". Thus the static activation flattening transfers the challenge of activation quantization to weights using the scaling $s$, but the term to compensate to this compensate weight error, $R$, is in low-precision in SERQ. I wonder whether the activation flattening is really helpful in this case considering MXFP has shared scales to accommodate outliers.
3. The GPU performance evaluation is very helpful. Could the author offer more details of the setup of GPU performance, like the batch size, token numbers, and tokens per second throughput?

---

> ### Author Response · Authors · 2025-11-21
> **Part 1**
>
> We really appreciate the time and thoughtful comments. We hope that the following response fully addresses the concerns raised by Reviewer cLFy.
>
> > **W1: The evaluation lacks model larger than 13B: all results were collected on model between 1B and 13B. It will be convincing if the author could offer results on larger models like 70B to verify the scalability of SERQ**
> ---
> We fully agree with your point that evaluating large-scale models over 13B is crucial to verify the scalability of our method. To address this, we conducted additional experiments using Llama-2-70B. We have incorporated these new results into **Table 1** of the revised manuscript. For your convenience, the results for the 70B model are presented below.
>
> | Model | Bits | Method | PPL | 0-shot | MMLU |
> | --- |  --- |  :--- |  --- |  --- |  --- |
> | LLaMA-2 70B | FP16 | baseline | 3.53 | 70.54 | 65.44 |
> | | W4A8 | LLM.int4() | 7.41e+3 | 49.71 | 43.56 |
> | | | L$^2$QER | 3.55 | 69.78 | 64.45 |
> | | | SERQ (RTN) | 3.44 | **70.35** | 64.45 |
> | | | SERQ (GPTQ) | **3.43** | 70.31 | **64.62**|
> | | W4A4 | LLM.int4() | 2.76e+4 | 39.77 | 23.67 |
> | | | L$^2$QER | 4.55 | 66.92 | 56.44 |
> | | | L$^2$QER-MXFP4 | 3.81 | 69.03 | 61.48 |
> | | | SERQ (RTN) | 3.65 | **69.68** | 63.15 |
> | | | SERQ (GPTQ) | **3.64** | 69.38 | 63.09 |
> | | | SERQ-MXFP4 | 3.79 | 69.58 | 61.64 |
>
> As shown in the results, our method consistently achieves state-of-the-art performance among LoRA-based quantization approaches, even on the 70B model. Thank you for your suggestion, which allowed us to further improve our manuscript.
>
> &nbsp;
>
> > **W2: Details of runtime/hardware performance needs further clarification. Could the author elaborate more how a SERQ layer is accelerated on GPU? are there kernel fusions?**
> ---
> We agree that the original manuscript lacked sufficient details regarding the GPU performance and implementation. First of all, Regarding how SERQ achieves acceleration on GPUs, our advantage comes from two main factors. (i) unlike existing LoRA-based quantization methods, SERQ constructs the low-rank path using a single narrow GEMM (pure 4-bit linear computation), which enables significant speedups. (ii) compared to online transform-based quantization techniques, our method introduces only the minor overhead of this low-rank adapter without requiring any additional computational layers within the model structure.
>
> In short, the only computational overheads in our framework are the narrow GEMM and the quantization kernel. Regarding your question on kernel fusion, since we utilize CUTLASS kernels, which are already highly optimized for GEMM operations on GPUs, we did not implement additional custom kernel fusion. This structural simplicity ensures that our method can be readily deployed with minimal modification, not only on GPUs but also various edge devices.
>
> &nbsp;
>
> > **Q1: What does the "Quarot" block in Figure 2(b) mean? Is this for rotating activations?**
> ---
> We sincerely apologize for the poor resolution of the figure, which led to this confusion. The text in the yellow block of **Figure 2(b)** is actually 'Quant' (short for Quantization), not 'Quarot'. It simply represents the online-quantization operation block.
> To address this visibility issue, we have updated the figure in the revised manuscript with larger and clearer text. We apologize for any misunderstanding this may have caused.
>
> &nbsp;
>
> > **Q2-1: SERQ is a heuristic based method. An ablation study of the static activation flattening seems needed here.**
> ---
> We appreciate your insightful comment. We fully acknowledge that our proposed method relies on heuristics. To provide a more robust theoretical grounding and mitigate concerns regarding its heuristic nature, we have added comprehensive theoretical analysis and supporting experiments in **Appendix A.4** and **Table 8**.
>
> Furthermore, since our approach combines Static Activation Flattening (SAF) with the SERQ mechanism, we completely agree with your point that an ablation study is necessary to isolate and evaluate the individual contributions of each component. Accordingly, we have conducted additional experiments and the results are as follow:
>
> | Bits | Q-config. | Method | LLaMA-2 7B | LLaMA-3.2 1B | LLaMA-3.2 3B | Qwen2.5 3B |
> | :---: | :---: | :--- | :---: | :---: | :---: | :---: |
> | FP16 | - | baseline | 5.47 | 9.75 | 7.81 | 8.03 |
> | W4A8 | INT (RTN) | only-SAF | 6.64 | 25.49 | 12.75 | 12.41 |
> |  |   | SERQ w/o SAF  | 5.64  |  11.13 | 8.37  | **8.56**  |
> |   |   | SERQ  | **5.63**  | **11.1** | **8.36**  | 8.57  |
> | W4A4 | INT (RTN) | only-SAF | 7.58 | 56.92 | 18.84 | 20.67 |
> |  |   | SERQ w/o SAF  | 6.05  |  13.6 |  **9.36** | 10.83  |
> |   |   | SERQ  | **6.03**  | **13.49** | 9.42  | **9.57** |
> |   | MXFP4 |  only-SAF | 7.03  | 17.21  | 11.49  | 12.88  |
> |   |   | SERQ w/o SAF  | **6.21**  | 13.56  | **9.59**  | 10.4  |
> |   |   |  SERQ   | 6.22  | **13.53**  | 9.69  | **9.84**  |
>
> We sincerely thank you for this constructive feedback.

---

> > ### Author Response · Authors · 2025-11-21
> > **Part 2**
> >
> > > **Q2-2: SmoothQuant was originally proposed for INT8 quantization, while SERQ uses MXFP.**
> > ---
> > We appreciate your accurate observation that SmoothQuant was originally designed for INT8 quantization. Nevertheless, we included it as a baseline because our method employs Static Activation Flattening (SAF), which shares similarities with the smoothing technique in SmoothQuant. Therefore,  we deemed it necessary to include it as a baseline to evaluate the effectiveness of our flattening approach (as also done in prior W4A4 works such as Quarot and SpinQuant). The 4-bit SmoothQuant results in **Table 2** were obtained by adapting the original algorithm to 4-bit precision.
> >
> > As for our initial choice of the MX-format, it was motivated by emerging hardware trends, such as the NVIDIA Blackwell architecture, which is prioritizing support for 4-bit floating-point formats over native 4-bit integers. While our main concepts are fundamentally rooted in integer quantization (SERQ-RTN and GPTQ are all INT-based), we aimed to align our evaluation with these hardware trends. However, to address your concern and ensure a strictly fair comparison, we have updated **Table 2** to unify all evaluations using the integer format.
> >
> > &nbsp;
> >
> > > **Q2-3: In line 262, "However, unlike standalone prior approaches, the combined use of low-rank reconstruction enables effective compensation for the induced weight errors.". Thus the static activation flattening transfers the challenge of activation quantization to weights using the scaling , but the term to compensate to this compensate weight error, , is in low-precision in SERQ. I wonder whether the activation flattening is really helpful in this case considering MXFP has shared scales to accommodate outliers.**
> > ---
> > Thank you for raising this insightful point. We agree that the MX-format is inherently robust to outliers because its fine-grained quantization (block size of 32) localizes the impact of activation outliers to small groups.
> > However, while MXFP4 effectively confines the scope of the outlier's influence, the magnitude of the outlier persists, still causing significant distortion within its specific group. To address your concern and verify this hypothesis, we conducted an additional ablation study in **Table 6** that includes comparisons with the MX-format.  The results are summarized in the table below.
> >
> > | Bits  | Method | LLaMA-2 7B | LLaMA-3.2 1B | LLaMA-3.2 3B | Qwen2.5 3B |
> > | :---: | :--- | --- | :---: | :---: | :---: |
> > | FP16 | baseline | 5.47 | 9.75 | 7.81 | 8.03 |
> > | MXFP4 |  only-SAF | 7.03  | 17.21  | 11.49  | 12.88  |
> > | | SERQ w/o SAF  | **6.21**  | 13.56  | **9.59**  | 10.4  |
> > | | SERQ   | 6.22  | **13.53**  | 9.69  | **9.84**  |
> >
> > As the results demonstrate, there is a noticeable performance drop in smaller models when SAF is not applied in SERQ. This is because, in smaller models, a partial sum from a single dequantized group has a relatively larger impact on the final output element due to the fewer number of accumulation terms. Consequently, to guarantee robust performance across all model scales, we identify SAF as an essential component of our approach.

---

> > > ### Author Response · Authors · 2025-11-21
> > > **Part 3**
> > >
> > > > **Q3: The GPU performance evaluation is very helpful. Could the author offer more details of the setup of GPU performance, like the batch size, token numbers, and tokens per second throughput?**
> > > ---
> > > We are really encouraged by your positive feedback regarding our GPU performance evaluation. And we agree that the specific parameters used in the experiment were not sufficiently detailed in the original manuscript. The results presented in **Figure 3** were measured using a batch size of 1 and a token length of 4k. To clarify the setup, We have updated the caption of **Figure 3** to explicitly include these specifications.
> > > Furthermore, regarding throughput, we have provided additional end-to-end inference speed mesurements in **Table 3**. The results are summarized below.
> > >
> > > | Bsz | Method | TTFT(ms) | Speedup | Peak Mem(GB) | Saving | TPOT(ms) | Speedup |
> > > | --- | :--- | --- | --- | --- | --- | --- | --- |
> > > | 1 | FP16 | 132.38 | $\times$1 | 17.44 |  $\times$1 | 18.44 | $\times$1 |
> > > | | MXFP4 | 56.03 | $\times$2.36 | 6.94 | $\times$2.51 | 25.48 | $\times$0.72 |
> > > | | SERQ-MXFP4 | 62.31 | $\times$2.12 | 7.03 | $\times$2.48 | 39.28 | $\times$0.47 |
> > > | 8 | FP16 | 1252.24 | $\times$1 | 27.9 |  $\times$1 | 61.93 | $\times$1 |
> > > | | MXFP4 | 542.58 | $\times$2.31 | 17.4 | $\times$1.6 | 54.49 | $\times$1.14 |
> > > | | SERQ-MXFP4 | 587.17 | $\times$2.13 | 17.48 | $\times$1.6 | 57.08 | $\times$1.08 |
> > > | 16 | FP16 | 2515.08 | $\times$1 | 39.85 |  $\times$1 | 114.24 | $\times$1 |
> > > | | MXFP4 | 1113.06 | $\times$2.26 | 29.35 | $\times$1.36 | 100.71 | $\times$1.13 |
> > > | | SERQ-MXFP4 | 1206.69 | $\times$2.08 | 29.44 | $\times$1.35 | 103.27 | $\times$1.11 |
> > > | 32 | FP16 | 4987.05 | $\times$1 | 63.76 |  $\times$1 | 218.01 | $\times$1 |
> > > | | MXFP4 | 2257.54 | $\times$2.21 | 53.25 | $\times$1.2 | 193.7 | $\times$1.13 |
> > > | | SERQ-MXFP4 | 2453.29 | $\times$2.03 | 53.34 | $\times$1.2 | 202.13 | $\times$1.08 |
> > >
> > > We hope this information satisfactorily addresses your question.

---

> > > > ### Comment · Reviewer_cLFy · 2025-11-27
> > > >
> > > > Thank you for clarification. Please add the additional experiments on number formats. I remain positive about this work.

---

> > > > > ### Author Response · Authors · 2025-12-01
> > > > >
> > > > > We sincerely thank you for taking the time to review our response. We would like to confirm that the experiments referenced in our rebuttal are fully included in the revised manuscript. Specifically, the experiments related to the format are presented in **Table 7**, with the corresponding explanation provided at **line 518**.
> > > > >
> > > > > We apologize for any confusion caused by initially showing only the MXFP4 portion of Table 7. Finally, we greatly appreciate your positive evaluation of our work.

---

### Official Review · Reviewer_emTw · 2025-10-31

**Soundness:** 2
**Presentation:** 3
**Contribution:** 2
**Rating:** 4
**Confidence:** 2

**Summary:**

This paper proposes a saliency-aware error reconstruction method named SERQ for LLMs in W4A4 using a single low-rank compensation matrix. The main contributions are threefold: static activation flattening, saliency-aware error reconstruction, and offline weight permutation. Experimental results show accuracy improvements with small computation overhead.

**Strengths:**

* Well-written with good presentation
* Systematic evaluation and good results

**Weaknesses:**

* Missing related work
* Missing theoretical justification

**Questions:**

* Ablation study: how much does static activation flattening affect your results?
* What are the design space of choosing saliency metrics? What do the authors mean by "sufficient"?
* Could the authors comment on how their work compares to QERA (ICLR 2025)?

---

> ### Author Response · Authors · 2025-11-21
> **Part 1**
>
> First of all, I would like to sincerely thank you for the time and effort you dedicated to reviewing our paper. I have done my best to address the concerns you raised in the following response.
>
> > **W1: Missing related work.**
> ---
> We originally intended to cover the related works in **Section 2.2**, but we acknowledge that the explanation was insufficient and the presentation was not really clear. We fully agree with your assessment. To address this, we have created a new section in the **Appendix A.2** to provide a comprehensive discussion of these studies. We would be grateful if you could refer to **Appendix A.2**. Thank you for this valuable comment.
>
> &nbsp;
>
> > **W2: Missing theoretical justification.**
> ---
> We fully acknowledge that the theoretical justification in the original manuscript was insufficient. As this concern is closely related to Question 2 (Q2), we have provided a detailed explanation in our response to that question. We kindly ask you to refer to our answer to Q2. Furthermore, we have conducted additional experiments to substantiate our theoretical justification, which can be found in **Appendix A.4**.
>
> &nbsp;
>
> > **Q1: Ablation study: how much does static activation flattening affect your results?**
> ---
> Thank you for the insightful question. As our method incorporates Static Activation Flattening (SAF) alongside Saliency-aware Low-rank Error Reconstruction (SERQ), we fully agree with your point that an ablation study is necessary to verify their individual contributions. Accordingly, we conducted additional experiments and updated **Section 4.3** with the results. Please refer to the table we added to the main paper below:
>
> | Bits | Q-config. | Method | LLaMA-2 7B | LLaMA-3.2 1B | LLaMA-3.2 3B | Qwen2.5 3B |
> | :---: | :---: | :--- | :---: | :---: | :---: | :---: |
> | FP16 | - | baseline | 5.47 | 9.75 | 7.81 | 8.03 |
> | W4A8 | INT (RTN) | only-SAF | 6.64 | 25.49 | 12.75 | 12.41 |
> |  |   | SERQ w/o SAF  | 5.64  |  11.13 | 8.37  | **8.56**  |
> |   |   | SERQ  | **5.63**  | **11.1** | **8.36**  | 8.57  |
> | W4A4 | INT (RTN) | only-SAF | 7.58 | 56.92 | 18.84 | 20.67 |
> |  |   | SERQ w/o SAF  | 6.05  |  13.6 |  **9.36** | 10.83  |
> |   |   | SERQ  | **6.03**  | **13.49** | 9.42  | **9.57** |
> |   | MXFP4 |  only-SAF | 7.03  | 17.21  | 11.49  | 12.88  |
> |   |   | SERQ w/o SAF  | **6.21**  | 13.56  | **9.59**  | 10.4  |
> |   |   |  SERQ   | 6.22  | **13.53**  | 9.69  | **9.84**  |
>
> As shown in the table, for normal-sized models, SERQ alone is highly effective to maintain performance without SAF. However, for smaller models, the benefits of SAF become much more pronounced. This is because the number of accumulated partial sums for a single output element is smaller, so a quantization error from a single group has an increased impact on the final output. We also present results for the MX-format, which is known to be relatively robust against outliers. Since the MX-format localizes the impact of outliers to small groups, not fundamentally resolves them, it exhibits a similar tendency to the INT-format. Consequently, to guarantee robust performance across all model scales, we identify SAF as an essential component of our approach.

---

> ### Author Response · Authors · 2025-11-21
> **Part 2**
>
> > **Q2: What are the design space of choosing saliency metrics? What do the authors mean by "sufficient"?**
> ---
> We fully agree that the explanation regarding the saliency metric in the original manuscript was insufficient, which may have made the theoretical justification appear lacking. We would like to take this opportunity to provide a detailed explanation of our design choices.
>
> * Regarding the design space of saliency metrics.
>
> Unlike conventional SVD-based quantization which uses singular values as the saliency metric, we selected the magnitude of the activation distribution as our criterion. While both approaches aim to minimize the error in a low-rank setting, their objectives differ slightly from ours. SVD-based methods perform truncated decomposition to minimize the quantization error of the weight matrix itself, i.e., $\min \| W - Q(W) \|$. However, in transformer models, the error propagates through linear operations ($Y = XW$), meaning the cumulative quantization error is effectively $\Delta Y = X \Delta W$. This leads to a crucial insight: minimizing $\Delta W$ globally is important, but it is far more critical to suppress $\Delta W$ specifically at indices where the input $X$ has large magnitudes. Therefore, we defined the magnitude of activation channels as our saliency metric and performed finer quantization on the corresponding weight rows to minimize the final output error. We empirically validated this in **Appendix A.4** and **Table 8**. The results are summarized below:
>
> | Model | Rank | Method | QSNR |
> | :---: | :---: | :--- | :---: |
> | LLaMA-3 8B | 128 | SVD | 19.87|
> | | | SERQ w/o SAF | 25.91 |
> | | 64 | SVD | 19.41 |
> | | | SERQ w/o SAF | 25.49 |
> | | 32 | SVD | 19.21 |
> | | | SERQ w/o SAF | 20.22 |
> | | 16 | SVD | 19.06 |
> | | | SERQ w/o SAF | 20.42 |
> | | 8 | SVD | 19.01 |
> | | | SERQ w/o SAF | 21.29 |
>
> &nbsp;
>
> * Regarding the term 'sufficient' (Line 242)
>
> We used this term to highlight the efficiency of our method compared to SVD. While SVD-based approaches typically require maintaining two low-rank matrices, which is suboptimal for latency, our method achieves superior performance with a single low-rank matrix, thanks to our saliency-aware strategy. We apologize if this phrasing caused any confusion.
>
> &nbsp;
>
> > **Q3: Could the authors comment on how their work compares to QERA (ICLR 2025)?**
> ---
> QERA is indeed a quantization study that utilizes low-rank error reconstruction, similar to SERQ and LQER. In particular, QERA introduces a closed-form solution that eliminates the  heuristics typically used to minimize linear layer output errors based on activation distributions, thereby enabling a more precise reconstruction of weight quantization errors.
>
> However, the primary distinction lies in the fact that QERA provides a mathematical framework only for weight quantization errors and does not address activation quantization. Since quantizing heavy-tailed activations is significantly more intricate than quantizing weights, which typically follow a Gaussian normal distribution, existing studies that quantize both weights and activations focus heavily on handling activation outliers.
>
> In this context, our work differentiates itself by addressing the difficulty of activation quantization through the two paths (full-weight/LoRA) of the LoRA architecture, while simultaneously optimizing the latency overhead in the low-rank path. Therefore, we excluded weight-only quantization results related to LoRA but instead included various SoTA rotation-based quantization work to prove our effectiveness. However, agree that discussing QERA provides valuable context as a related work, so we have incorporated this comparison into **Appendix A.2** in revised manuscript.
>
> &nbsp;
>
> Thank you once again for your valuable review. We are happy to engage in further discussion.
>
> Sincerely, The Authors

---

### Author Response · Authors · 2025-12-01
**Summary of Major Updates and Contributions for the Area Chair**

We would like to express our deepest gratitude to all reviewers for their insightful feedback. We have thoroughly addressed all reviewer comments and substantially improved the manuscript. For clarity, all major revisions in the updated version are highlighted in blue. We are pleased to report that we have successfully resolved all concerns raised by the reviewers who provided follow-up feedback. As a result, two reviewers responded positively to our rebuttal, with one indicating that they will re-evaluate their score. Although two reviewers have not yet assessed the revised manuscript, we have carefully addressed all of their points, including the additional experiments they requested. Taking these circumstances into account, we would greatly appreciate your positive consideration of our revised manuscript.

We are aware that recent technical issues on OpenReview may have increased the burden on the Area Chair. We deeply appreciate your dedication to managing this process under such circumstances. To assist in your decision-making process, we provide a summary of our key contributions and the major updates made during the rebuttal period.


> **1. Summary of Contributions**
---

* We proposed a novel quantization method, Saliency-aware Low-rank Error Reconstruction (**SERQ**), which successfully introduces a W4A4 scheme to LoRA-based quantization by jointly addressing both activation and weight quantization.
* We identified and resolved critical latency bottlenecks in LoRA-based quantization by maintaining a single low-rank matrix in the low-rank path, enabled by SERQ. This structural optimization reduces latency overhead to a negligible level compared to pure MXFP4 computation.

&nbsp;

> **2. Major Updates and Clarifications**
---
Based on the constructive comments from the reviewers, we have made the following major updates:

- **Scalability (Reviewers `cLFy`, `oK6e`):** We addressed the need for validation on large-scale models by adding experiments on LLaMA-2 70B. The results, incorporated into the main **Table 1**, confirm that our method satisfies the scalability requirements intrinsic to deep learning. As a result, SERQ consistently outperforms prior arts across both W4A8 and W4A4 quantization schemes.
- **End-to-End Performance (Reviewers `cLFy`, `3ZBf`):** We conducted additional experiments on end-to-end performance(prefilling and decoding speed) using the latest NVIDIA Blackwell GPUs. The results, added to **Table 3**, demonstrate speed-ups of over 2$\times$, closely mirroring the performance of pure MXFP4 while delivering significant memory savings. This effectively proves the real-world viability of our 4-bit implementation.
- **Theoretical Clarification (Reviewers `emTw`, `oK6e`):** we addressed concerns regarding theoretical justification by clarifying the design space of our saliency metric and the detailed mechanism behind SERQ's superior performance compared to prior works in **Appendix A.4**. We also added **Table 8** to demonstrate empirically that our quantization-aware saliency is more effective than traditional SVD.
- **Ablation Study on SAF (Reviewers `emTw`, `cLFy`):** We also recognized the importance of verifying the individual impact of Static Activation Flattening (SAF). Accordingly, we conducted additional experiments, now presented in **Table 7**. The results clearly indicate that the influence of SAF is particularly pronounced in smaller models. Consequently, we conclude that SAF is an indispensable component for ensuring performance stability in small-scale architectures.
- **Related Works (Reviewers `emTw`, `3ZBf`):** We acknowledge that Section 2.2 focused exclusively on low-rank error reconstruction, which resulted in insufficient coverage of other similar methods. To clarify this, we have expanded the discussion on similar works in **Appendix A.2**, explicitly highlighting the distinctiveness and novelty of our research compared to prior arts.

---
In summary, we have incorporated the reviewers’ feedback to significantly improve the quality and robustness of our manuscript. Once again, we extend our deepest gratitude to the AC for your hard work and dedication.

Sincerely, The Authors

---

### Meta-Review · Area_Chair_UQj2 · 2026-01-07

**Summary:**

This paper proposes SERQ (Saliency‑Aware Error Reconstruction for Quantization), a post‑training W4A4 (4‑bit weight, 4‑bit activation) quantization method for large language models that aims to achieve true end‑to‑end 4‑bit inference with minimal latency overhead. The key idea is to use a single low‑rank compensation matrix to correct quantization errors only for salient weight rows.

**Reviewer Concerns:**

**Theoretical Justification**

Early reviewers (emTw, oK6e) criticized the lack of theory behind:
* The saliency metric (activation magnitude).
* Why a single low‑rank matrix is sufficient.

Authors addressed this by:
Adding  theoretical analysis and demonstrating empirically that saliency‑aware row selection outperforms SVD‑based criteria when the low‑rank path itself is quantized.

**Ablation**

Reviewers requested clear ablations for Static Activation Flattening (SAF).
Authors added detailed ablations showing:
SERQ works well without SAF on large models.
SAF is critical for smaller models and MXFP formats where partial sums are fewer and errors amplify.
Fairness of Comparisons / Bit‑Budget


**Concerns about**
- Lack of large‑scale models (70B+).
- Missing long‑context evaluation.
- No MoE analysis.
Authors addressed:
Added LLaMA‑2‑70B results.
Highlighted existing LongBench (4K–32K) results.
Provided architectural arguments for MoE compatibility, with experiments stated as ongoing.


Reviewers questioned:
Whether offline permutation breaks tensor parallelism.
Calibration cost of SERQ as a PTQ method.
Authors added:
Calibration time comparisons showing SERQ is among the fastest PTQ methods (~23 minutes vs hours for others).
Clarified SERQ adds only a single narrow GEMM and no online transforms.

**Reviewer Scores:**

cLFy (Score: 8 → strong accept)
Very positive; asked for larger models and hardware details. After rebuttal and added 70B + throughput experiments, explicitly remained positive.

oK6e (Score: 4 → reconsidering)
Initially very critical on fairness, theory, and scope. After extensive rebuttal and added experiments, explicitly stated they would reconsider their score.

3ZBf (Score: 4)
Borderline; main concerns were missing recent baselines and unfair format comparison—both addressed.

emTw (Score: 4)
Initial concerns on missing related work and theory were directly addressed with new appendix sections and ablations.

Given the trend, I suggest an accpet.

---

### Decision · Program_Chairs · 2026-01-26

Accept (Poster)